# Flexi-YOLO: A lightweight method for road crack detection in complex environments

**Jiexiang Yang**[1]*, **Renjie Tian**[2], **Zexing Zhou**[1], **Xingyue Tan**[3], **Pingyang He**[1]

**1** School of Information Science and Engineering, Chongqing Jiaotong University, Chongqing, China, **2** Institute of Computer Vision and Traffic Image Understanding, School of Information Science and Engineering, Chongqing Jiaotong University, Chongqing, China, **3** School of Mathematics and Statistics, Chongqing Jiaotong University, Chongqing, China

☯ These authors contributed equally to this work.
\* 632207100225@mails.cqjtu.edu.cn

**Data availability statement:** The data supporting the findings of this study are openly available in Zenodo at https://doi.org/10.5281/zenodo.15019275.

## Abstract

Road crack detection is critical to global infrastructure maintenance and public safety, and complex background environments and nonlinear damage crack patterns challenge the need for real-time, efficient, and accurate detection.This paper proposes a lightweight yet robust Flexi-YOLO model based on the YOLOv8 algorithm. We designed Wise-IoU as the model's loss function to optimize the regression accuracy of its bounding boxes and enhance robustness to low-quality samples. The DCNv-C2f module is constructed for the transformation and fusion of feature information, allowing the convolutional kernels to adapt to the complex shape characteristics of cracks dynamically. A Global Attention Module (GAM) is integrated to improve the model's perception of global information. The AKConv convolution operation is employed to adaptively adjust the size of convolutions, further enhancing local feature capturing. Additionally, a lightweight network design is implemented, establishing G-Head (Ghost-Head) as the detection head to optimize the issue of feature redundancy. Experimental results show that Flexi-YOLO achieves an accuracy increase of 2.7% over YOLOv8n, a recall rate rise of 4.7%, a mAP improvement of 5.3%, a mAP@0.5–0.95 increase of 3.9%, a decrease of 0.5 in GFLOPS, and an F1 score improvement from 0.80 to 0.84. Flexi-YOLO offers higher detection accuracy and robustness and meets the industrial demands for lightweight real-time detection and lower application costs, providing an efficient and precise solution for the automated detection of road cracks.

## Introduction

Roads are a critical component of global infrastructure and are essential for the long-term stable development of the world economy. However, due to prolonged use and various environmental factors, infrastructure structures inevitably experience aging, with concrete structures particularly susceptible to cracking [1]. These cracks affect road durability, safety, and usability, reduce their load-bearing capacity, shorten their lifespan, and even threaten user safety [2]. Structural Health Monitoring (SHM) has expanded into diverse fields, including

**Funding:** This work was supported by the National Natural Science Foundation of China under Grants (62276034, 62306052), Group Building Scientific Innovation Project for Universities in Chongqing (CXQT21021), Joint Training Base Construction Project for Graduate Students in Chongqing (JDLHPYJD2021016), and Science and Technology Research Program of Chongqing Municipal Education Commission (KJQN202100712). The sponsor has not participated in research design, data collection and analysis, decision to publish or write manuscripts, and the sponsor only supports the hardware equipment and computing support of the experiment and the payment of publishing layout fees.

**Competing interests:** The authors have declared that no competing interests exist.

bridges, pipelines, tunnels, and buildings. However, limitations remain in Structural Damage Identification (SDI) and precise crack detection and analysis. Sarkar et al. [3] address this gap by exploring the development and application of Artificial Intelligence in civil engineering, focusing on concrete structures. Their research on concrete provides methods and conclusions applicable not only to road crack detection but also to the health monitoring of other infrastructure.According to the World Health Organization's 2023 report, approximately 1.19 million people die each year due to road traffic accidents, with nearly 3,400 deaths per day and up to 50 million injured@0.5. Among these, individuals aged 5 to 29 represent the primary age group for road accident fatalities, which is also a crucial demographic for nations' current and future economic development, bearing the responsibility for significant economic construction.

If road cracks are not maintained or repaired, minor cracks can develop into deeper fissures and potholes, leading to greater economic losses. As one of the primary materials used in infrastructure such as roads, bridges, and tunnels, the structural health of concrete is a significant concern for these facilities. Enhancing concrete's inherent strength and durability to improve the safety of foundational structures is a reliable approach [4]. Mandal et al. [5] utilized machine learning techniques to predict the performance of concrete based on the mixing ratios of its components, while Shiuly et al. [6] applied various machine learning methods to forecast the compressive strength of mortar and concrete. Their research provides cost-effective preventive measures for infrastructure safety from the material perspective. Crack detection is a key indicator for assessing the condition of concrete structures and monitoring the safety and maintainability of infrastructure. Employing computer vision for non-destructive testing (NDT) of cracks is another mainstream method [3], which plays a crucial role in subsequent road maintenance and repairs. Therefore, timely detection and assessment of road cracks are necessary to reduce later repair and maintenance costs [7].

Furthermore, the expansion and deterioration of road cracks can negatively impact the surrounding environment and ecosystems. Some plants may take root in the cracks, further widening them, while water accumulation in the cracks can lead to soil erosion and vegetation damage, disrupting ecological balance. Debris and harmful substances in the cracks may also seep into groundwater resources, contaminating water quality and threatening public health [8]. Thus, road crack detection is a technical issue and a significant concern related to public safety, environmental protection, and sustainable development. There is an urgent need to research and develop more efficient and accurate low-cost crack detection technologies to address the increasingly severe problem of road cracks.

In the field of crack detection, it is usually divided into two stages. The first stage involves using traditional methods for detection. However, with the advancement of artificial intelligence, methods utilizing computer vision are gradually replacing traditional techniques. Although mainstream deep learning methods in road crack detection have made significant progress, they still face several key challenges. Factors such as background noise, surface texture, lighting shadows, crack edges, and variations in crack shape, size, depth, and moisture often lead to false positives and missed detections, increasing the difficulty of detection [9]. These factors require models to not only adapt to different background noises but also possess the ability to handle surface textures and shadows, enhancing their resistance to interference. Road safety work is often conducted on mobile devices that typically have limited computational power and lower application costs. Therefore, it is necessary to ensure that the models are lightweight and flexible enough to adapt to such devices.

In response to the aforementioned issues, this paper proposes the Flexi-YOLO model, which is based on the YOLOv8 algorithm and tailored explicitly for road crack detection,

effectively addressing current challenges in identifying road cracks. This work aims to overcome the difficulties in road crack detection by achieving the following objectives:

Develop a lightweight model suitable for deployment on devices with limited computational resources, design a backbone network that enhances the capability to extract features of complex geometrical shapes of cracks, integrate attention mechanisms and loss functions to improve the model's detection accuracy and robustness, and redesign a detection head to reduce feature redundancy and computational load, ensuring the model's lightweight nature to meet industrial requirements.

The following are the main points of discussion in this article:

**1**. Introducing variable convolution DCNv3 in the C2f module enhances the feature extraction of morphologically diverse cracks, effectively solving the difficult detection problem caused by the specificity of crack features (length is much larger than width).

**2**. Replaced part of the convolution operations with variable kernel convolution AKConv, redesigned the initial sampling points, and utilized the adjustable characteristics of variable convolution kernels to adapt dynamically to the complex geometrical shapes of cracks, strengthening the representation of local features and the fusion capabilities of the backbone network's features.

**3**. Fusing the GAM attention mechanism in the neck network, utilizing its powerful global information extraction capability, effectively retains the image detail information and enhances the relevance of the global information, thus reducing the leakage and misdetection rates of the model.

**4**. Design a new lightweight detection head G-head, reduce the redundant features in the model, reduce the computational complexity of the model while ensuring the accuracy, so that the model can be adapted to low-computing-power devices.

**5**. Utilized the Wise-IoU loss function to optimize boundary box regression, enhancing the model's adjustment to low-quality sample parameters and improving its robustness and accuracy in processing boundary boxes.

## Related work

### Traditional methods

Manual inspection and visual examination in crack detection remain the primary methods in most developing countries. However, manual inspection faces a series of issues, including but not limited to low efficiency, subjective judgment factors affecting assessments, difficulty in detecting fine cracks with the naked eye, high labor costs, and significant errors [10]. Thermal imaging technology has also garnered attention for crack detection due to its high portability and insensitivity to lighting conditions, offering specific advantages; however, this method is quite expensive, and its resolution needs improvement [11]. Ultraviolet and laser techniques [10,12] are frequently used in the detection of pavement cracks, as they can produce high-resolution images. However, this approach requires complex equipment and has high costs. Ground Penetrating Radar (GPR) is a non-destructive testing method widely used for pavement inspection [13,14], but it generates large amounts of data, has slow processing times, and suffers from subjective data interpretation and inconsistent standards. While these traditional techniques each have advantages, none have successfully balanced the issues of accuracy, efficiency, and cost.

In recent years, with the rapid development of computer vision, researchers have provided new technological directions for crack detection, which can be broadly categorized into two types. The first type is based on traditional digital image processing, primarily involving manual feature discrimination, where specific feature recognition criteria are designed

to facilitate identification [15–17]. Kim J T et al. [18] utilized structural frequency changes to detect and locate cracks, while Nguyen et al. [19] employed the geometric properties of cracks in images to identify their edges. Jin et al. [20] proposed a method for detecting pavement cracks by integrating directional gradient histograms with the watershed algorithm. However, traditional image processing methods struggle to meet crack detection's accuracy and time requirements [21]. It is noteworthy that while traditional crack detection methods achieved some success compared to manual inspections in earlier stages, their heavy reliance on human intervention and fixed detection scenarios has led to persistent issues such as inefficiency and poor generalization performance, prompting researchers to explore alternative approaches. The second type is based on deep learning theory, utilizing convolutional neural networks for autonomous feature learning, thereby achieving the task of crack detection. Due to its high levels of automation, intelligence, precision, and robustness, this approach has become the mainstream method in the detection field and is widely adopted by scholars [22–24].

## Deep learning methods

Deep learning methods have addressed some issues of traditional crack detection techniques, reducing detection errors caused by human subjectivity and overcoming the limitations of conventional image processing methods in recognizing complex backgrounds in practical engineering applications.

Deep learning-based detection can be broadly classified into two categories: two-stage detection algorithms and single-stage detection algorithms. The former first generates a series of candidate regions and then classifies these regions and performs bounding box regression. Representative models include R-CNN, Fast R-CNN, Faster R-CNN, and Mask R-CNN [25–28]. The latter directly predicts classes and bounding boxes on the image without the need to generate candidate regions, with models such as You Only Look Once (YOLO) and Single Shot MultiBox Detector(SSD) being representative [29,30]. However, existing deep learning methods still face numerous challenges, as most models have high computational complexity and slow processing speeds, which limit their application scenarios [31]. Many models are sensitive to environmental noise and prone to false positives in noisy backgrounds, and there is limited exploration of application scenarios across different domains. To address these issues, researchers have proposed various optimization strategies. Shiuly et al. [6] compared the effectiveness of six models in predicting the compressive strength of concrete, demonstrating the efficacy of deep learning models in predicting material performance and providing new insights for the application of deep learning in structural performance assessment. Cha et al. [32] used CNN in combination with the sliding window technique to scan the test images to make the model excel at detecting thin cracks under illumination conditions. Hacıefendioğlu et al. [33] used Faster R-CNN to study crack detection to conclude that lighting conditions have the highest impact on crack detection. Xu et al. [34] found that the former is better than the latter under certain conditions by comparing the detection of cracks with Faster R-CNN and Mask R-CNN, but the two algorithms require large and complex datasets. To address these problems, researchers have proposed various optimization strategies. For example, Zhang et al. [35] introduced MobileNetV2 and CBAM into YOLOv3 to reduce the network parameters. Yu et al. [36] pruned YOLOv4 to significantly increase the detection rate while ensuring the accuracy to contribute to the UAV crack detection. Chen et al. [37] introduced the Ghost module and CA attention mechanism in YOLOv5 for the backbone network to effectively improve the detection of coal cracks, which contributes to the construction of intelligent mines. Chen et al. [38] proposed a sample enhancement strategy based on cycle-GAN, which uses a migration learning strategy to incorporate the Transformer

attention mechanism in YOLO v5 to help the detection of cracks by drones. Transformer attention mechanism, to help the network find the region of interest in the complex background, to improve the detection and identification of small-scale defects, for the pipeline defects task to solve the overfitting problem brought by small samples and large models. Tran et al. [39] compared the mainstream models, selected YOLOv7, and proposed a U-Net model that can detect and segment bridge deck cracks in a fast and high-precision way. detection and segmentation of bridge surface cracks. Xiong et al. [40] incorporated the GAM attention mechanism and Wise-IoU into YOLOv8 to study the bridge surface cracks and found that the GAM attention mechanism has a great effect on crack detection. Wang et al. [41] optimized YOLOv8 with simSPPF using the spatial pyramid pooling layer in YOLOv8 and introduced the dynamic large convolutional kernel LSK attention mechanism, making the model more effective for pavement defect detection. Liu et al. . [42] introduced the C2f_AK module in the neck network to enhance feature fusion, significantly improved the detection head, and reduced parameters to achieve model lightweighting.

It is noteworthy that YOLO has not only achieved results in the field of crack detection but also in agricultural applications. Di et al. [43] proposed the DF-Tiny-YOLO model for apple leaf disease detection, effectively addressing the issue of distinguishing between healthy and diseased apple leaves. Liu et al. [44] introduced the ETSR-YOLO model to tackle the challenge of traffic sign recognition in road environments, providing a new solution for traffic sign recognition on embedded platforms in vehicles. Gong et al. [45] developed a two-stage ship detection method based on YOLOv8 for long-distance vessel detection, offering new insights into the field. Xu et al. [46] designed three plug-and-play modules—DGLFG, SCCFF, and ISPP—based on YOLOv8, significantly improving the feature learning capabilities for detecting small objects from aerial photography.

Sapkota et al. [47] found that, when compared to YOLOv8, YOLOv9, and YOLOv10, YOLOv8 maintains accuracy and inference speed while achieving a lower computational cost. However, despite its strong performance in multi-scale feature extraction, YOLOv8 still struggles with detecting small objects and defect cracks, leading to frequent missed detections and false positives. The primary issue is that as the network layers increase, some shallow feature information becomes difficult to retain [48].

One-stage algorithms provide faster detection speeds in real-time scenarios but sacrifice some level of accuracy. The optimization efforts in the studies mentioned above mainly focus on the backbone and neck networks, overlooking the potential of the detection head in model optimization. While these works have achieved remarkable outcomes, there are still deficiencies in balancing the model for detection tasks in non-ideal backgrounds and the pursuit of lightweight solutions. To address the limitations of deep learning methods in lightweight crack detection tasks under complex backgrounds, we aim to enhance the model's feature extraction and fusion capabilities by redesigning the backbone and neck network structures. Additionally, we propose an innovative detection head to fill the gap in optimizing the detection head to achieve model lightweightness, thereby catering to the adaptability of edge computing or mobile infrastructures.

## Method

### YOLOv8 algorithm

The YOLOv8 algorithm is a fast single-stage object detection method. It primarily consists of three key components: the backbone network, the neck network, and the prediction output head.

The backbone network forms the core of YOLOv8, based on the CSP (Cross Stage Partial) concept. It features a relatively simple structure akin to the traditional C3 module, as shown in Fig 1, employing the CSPNet idea of feature partitioning while integrating a residual structure. This allows for the stacking of n-Bottleneck modules to extract features efficiently. Compared to the C3 module, YOLOv8 incorporates the more richly flowing gradient C2f module, depicted in Fig 2. C2f draws inspiration from the ELAN concept introduced in YOLOv7 [49], enabling YOLOv8 to maintain a lightweight configuration while acquiring abundant gradient flow information. This facilitates the fusion of high-level features and contextual information, allowing the model to utilize detail and semantic information across different scales. At the end of the backbone network, the SPPF module is still utilized, sequentially using three max-pooling layers and then concatenating the outputs from each layer.

The neck section typically merges features from various stages of the backbone network to enhance the model's expressive capability. YOLOv8 continues to adopt the PAN-FPN (Path Aggregation Network with Feature Pyramid Network) paradigm. The top-down path established via FPN allows for the preliminary merging of multi-scale features, while the bottom-up path introduced through PAN facilitates the transfer of low-level features to high-level features, further enriching the multi-scale representation.

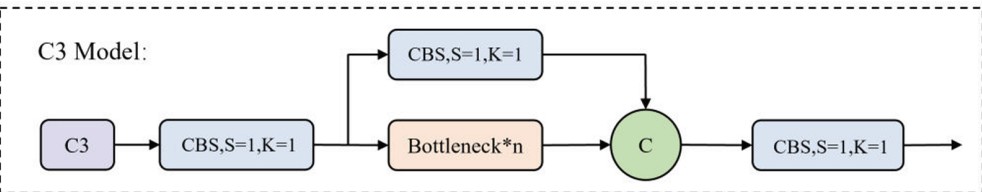

**Fig 1. C3 Module.**

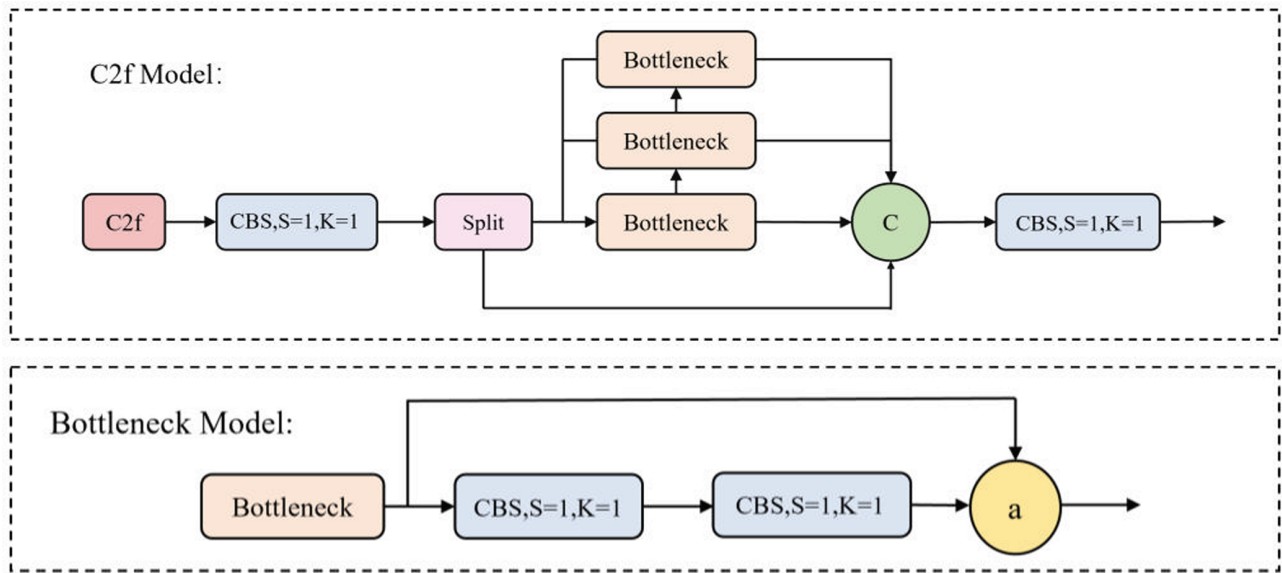

**Fig 2. C2f Module.**

The head section employs a decoupled design, separating the classification head from the detection head. It primarily processes the output of the features from the neck section, utilizing two 3x3 convolutions and one 1x1 convolution to calculate the Box loss and Class loss, ultimately yielding the final prediction results.

Despite the exceptional performance of YOLOv8, effectively handling occluded objects and cluttered scenes and adapting to overlapping or various crack shapes remains a long-term challenge for the YOLO family [50].

## Fleix-YOLO model

To address the difficulties YOLOv8 faces in accurately detecting, classifying, and localizing elongated targets (where the length of the object significantly exceeds its width) affected by environmental factors and background noise (such as lighting, contrast, and watermarks), as well as the challenges of adapting to low-computation devices, we propose a model called Fleix-YOLO, illustrated in Fig 3. This model initially tackles the detection and localization issues caused by the intrinsic elongated characteristics of the target by incorporating the concept of Deformable Convolution (DCNv3) from [51]. The integration of Adaptive Kernel Convolution (AKConv) from [52] allows the convolution kernel to adjust its size flexibly based on application requirements, further alleviating the detection and localization challenges posed by target features. To connect global information and retain more features, we introduce the GAM attention mechanism from [53], which aids in extracting important features from high-interference backgrounds, addressing the inadequate extraction of feature information.However, the global attention mechanism's capability to preserve feature details significantly increases model parameters and computational demands. To mitigate the model's weight, we have designed an innovative detection head that leverages the low computational complexity and effective feature retention of GhostConv [54], ensuring that model performance is not compromised while reducing computational load and parameters. Recognizing that datasets inevitably contain some low-quality annotations, we replace the original CIoU loss function with Wise-IoU [55], employing a dynamic non-monotonic focusing mechanism to evaluate anchor box quality using "outlier degrees," thereby lessening the adverse impact of low-quality annotations on model performance.

**Adaptive dynamic feature extraction backbone.** Feature extraction is a crucial step in model training. We have redesigned the backbone network to enhance the model's detection performance by extracting crack features at different scales and capturing more details related to crack features. This led to the development of the DCNv-C2f module and the AKConv module, which serve as core components of the backbone network.Traditional convolution typically utilizes fixed-size kernels (e.g., 1×1, 3×3, 5×5) for sampling input feature maps. However, this method often results in suboptimal feature extraction for irregularly shaped targets, exhibiting poor adaptability to unknown variations and limited generalization ability. Fixed-form convolutions have certain limitations when dealing with images that present complex and varied features [56]. Given the diversity of crack shapes and the interference of backgrounds, traditional convolution operations fail to meet the performance requirements for crack detection. The DCNv-C2f module introduces offsets at each sampling point to achieve dynamically adaptive receptive fields. This mechanism allows each convolution operation to extract features that more closely approximate the actual shape and size of the target features, overcoming the sampling limitations of traditional convolution with fixed geometric shapes. Furthermore, the powerful modulation mechanism of DCNv enhances the network's ability to focus on target regions, enabling the capture of nonlinear deformation features, particularly various irregular cracks. Under this design, the C2f layer generates outputs with greater

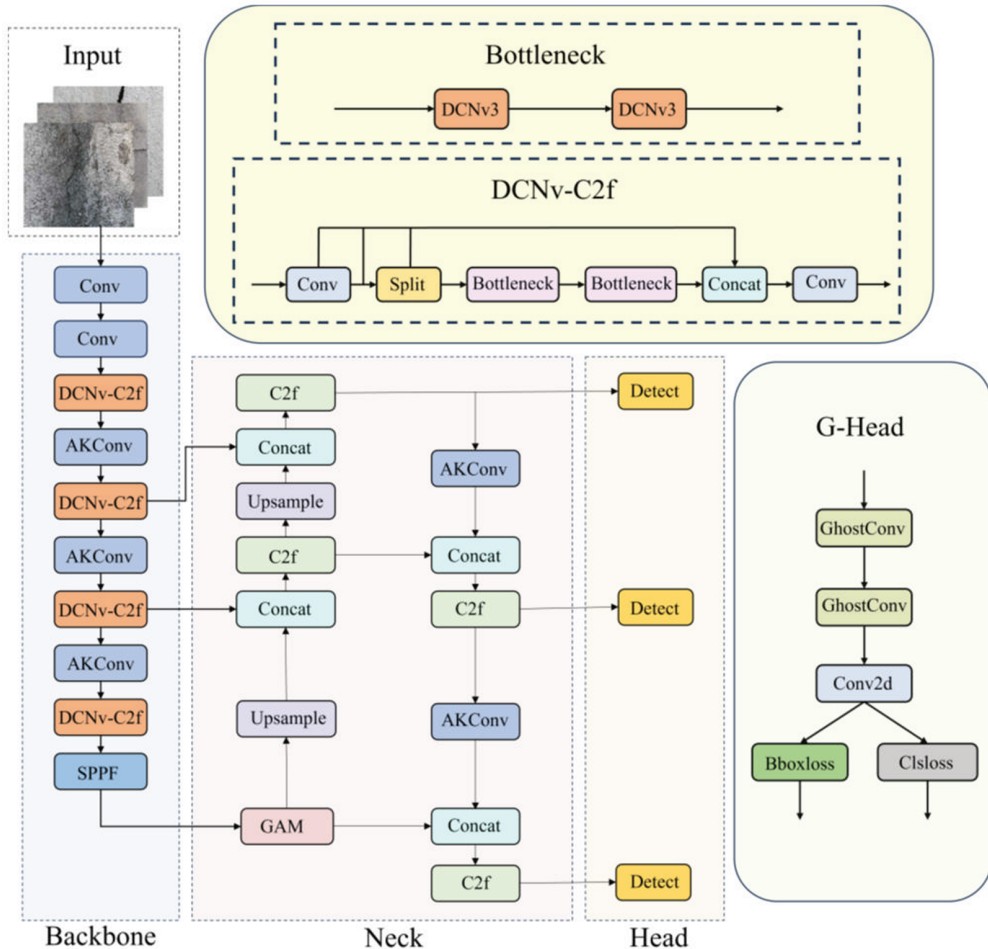

**Fig 3. Flexi-YOLO structure.**

representational power during each feature extraction, which helps improve the network's performance and expressive capability, making the feature extraction process more attuned to the characteristics of crack data, thereby significantly enhancing the model's detection performance. As shown in Fig 4, deformable convolution adds a directional parameter to each element of the kernel, allowing the position of sampling points to be dynamically adjusted based on the input content of the image, enabling the kernel to cover a larger area.

Feature matrix obtained by standard convolution:

$$y(P_0) = \sum_{P_n \in \mathfrak{R}} W(P_n) \cdot X(P_0 + P_n), \tag{1}$$

Where: $X$ represents the input feature map; $y$ denotes the output feature map; $P_0$ corresponds to the coordinates of real pixels in $y$; $P_n$ indicates the position of the convolution kernel; $W(P_n)$ refers to the weight of the nth sampling point; $X(P_0 + P_n)$ represents an element in the input feature map at position $P_0 + P_n$; $\mathfrak{R}$ is the regular network of the convolution kernel, and $\mathfrak{R} = \{(-1,-1),(-1,0),\cdots,(0,1),(1,1)\}$.

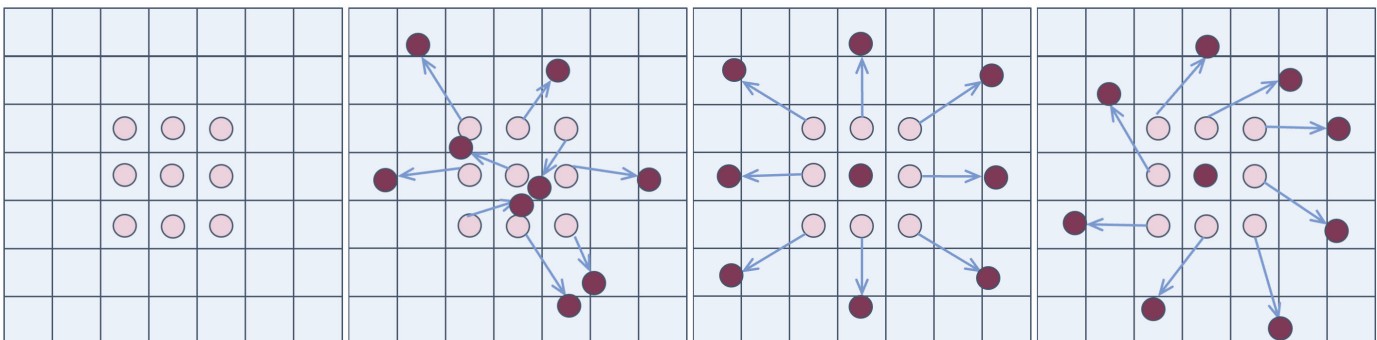

**Fig 4. Variable convolution.** illustrates the differences in sampling methods between standard convolution kernels and deformable convolution kernels using a 3x3 convolution kernel as an example. It describes the fixed rectangular pattern used during standard convolution sampling, the sampling pattern that changes based on offsets, the pattern involving scaling transformations to accommodate size changes, and the special sampling pattern formed by rotation.

The feature matrix obtained from DCNv3 [51]:

$$y(P_0) = \sum_{g=1}^{G} \sum_{k=1}^{K} w_g m_{gk} x_g(P_0 + P_k + \Delta P_{gk}) \qquad (2)$$

where: $G$ represents the number of aggregated groups; $w_g$ is the weight of the g-th group; $m_{gk}$ is the modulation scalar of the k-th sampling point in the g-th group; $\Delta P_{gk}$ is the offset relative to point $P_k$ in the base network [57].

In DCNv1 [58], only offsets were added to the sampling points of standard convolutions. Building on this, DCNv2 [59] allows control over the contribution of each sampling point to the output feature map. DCNv3 (as shown in Fig 5) incorporates the idea of separable convolutions, dividing the original convolution weights into two parts: depth-wise and point-wise. The point-wise component serves as shared projection weights between the sampling points to enhance overall model efficiency and reduce computational complexity. The depth component is handled by $m_{gk}$, while the point component is managed by W. A multi-group mechanism is introduced, segmenting the spatial aggregation process into $G$ groups, with each group having separate sampling offset $\Delta P_{gk}$ sand an adjustment scale $m_{gk}$ to achieve precise extraction of local features in cracks. Finally, the normalization function in DCNv2 is changed from Sigmoid to Softmax, constraining the sum of the modulation scalars to 1, which stabilizes the training process across different scales.

To further extract the subtle features of narrow cracks, AKConv is closely integrated with DCNv-C2f. After each DCNv-C2f operation, an AKConv convolution operation is added. A key feature of AKConv is that the size and shape of its adaptive convolution kernel can be dynamically adjusted according to actual needs [52,60]. This design reduces the original parameters from a quadratic growth to a linear growth, significantly decreasing the computational load; the structure of this module is shown in Fig 6. After the DCNv-C2f layer completes the initial feature representation, AKConv carries out feature fusion. AKConv dynamically generates convolutional kernel weights, adapting the convolution parameters based on the context of the input feature map. This mechanism enhances the model's ability to express multi-scale features and suppresses background noise. The backbone network aims to retain as much crack detail information as possible, with the collaborative enhancement mechanism of DCNv-C2f and AKConv, where the former handles local convolution modeling and the latter is responsible for multi-scale feature fusion. They share parameters and complement each

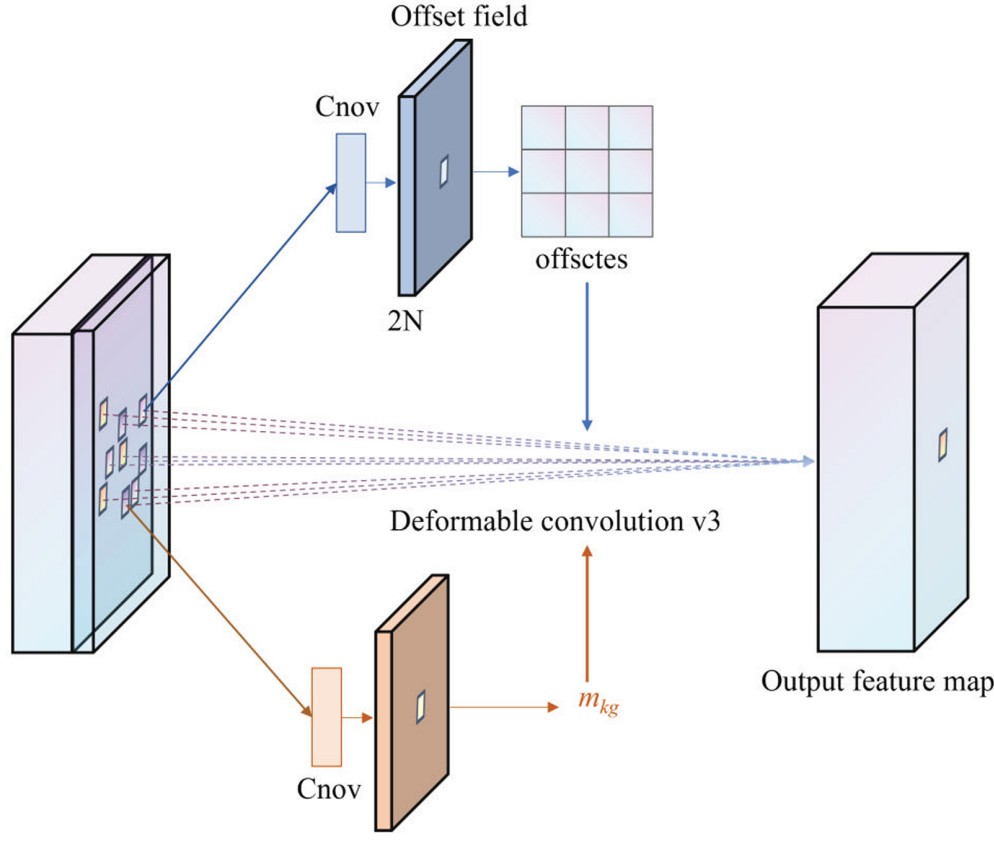

**Fig 5. Structure diagram of DCNv3.**

other, jointly improving the model's ability to extract complex geometric features of cracks. Additionally, AKConv utilizes SiLu as the activation function [61,62], enabling the model to learn more complex feature representations, adapt to variations in local features, and handle crack images in occluded and multi-interfering background environments.

Feature matrix of AKConv:

$$offset = Conv2d(X) \tag{3}$$

$$P_{new} = P_0 + P_n \tag{4}$$

$$\Gamma = Resample(\Gamma_{in}, P_{new}) \tag{5}$$

where: *Conv2d* represents the convolution operation, *X* denotes the input feature map, Resample refers to the interpolation and resampling operation, $P_n$ indicates the offset, $P_0$ signifies the initial coordinates, and $P_{new}$ represents the new coordinates.

In crack detection, AKConv can effectively adapt to pavement cracks of varying sizes and shapes while requiring less hardware equipment. The shape of cracks is typically elongated and longitudinally extending, necessitating dense longitudinal sampling to capture the details and direction of the cracks. To achieve this, we have optimized the initial sampling shape specifically for crack detection, as shown in Fig 7. By reducing the density of lateral sampling, we minimize unnecessary lateral sampling points, enhance the efficiency of feature capture,

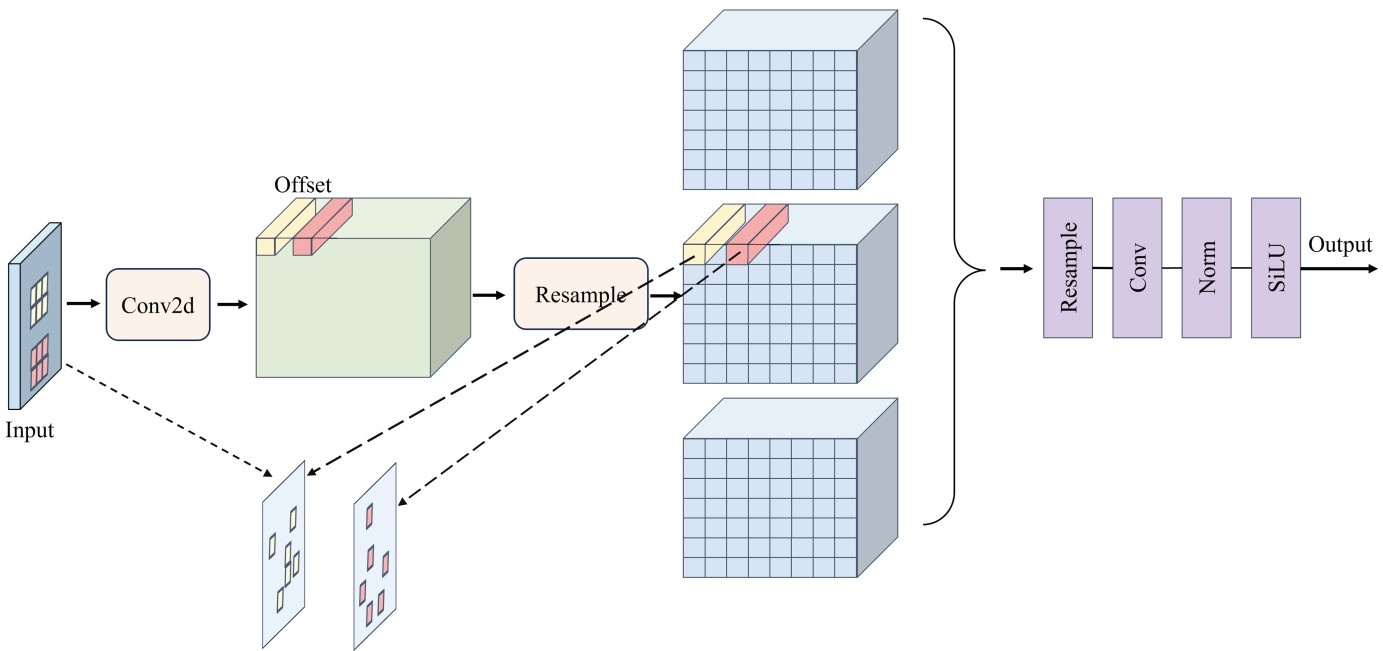

**Fig 6. Variable convolution Kernel structure diagram.**

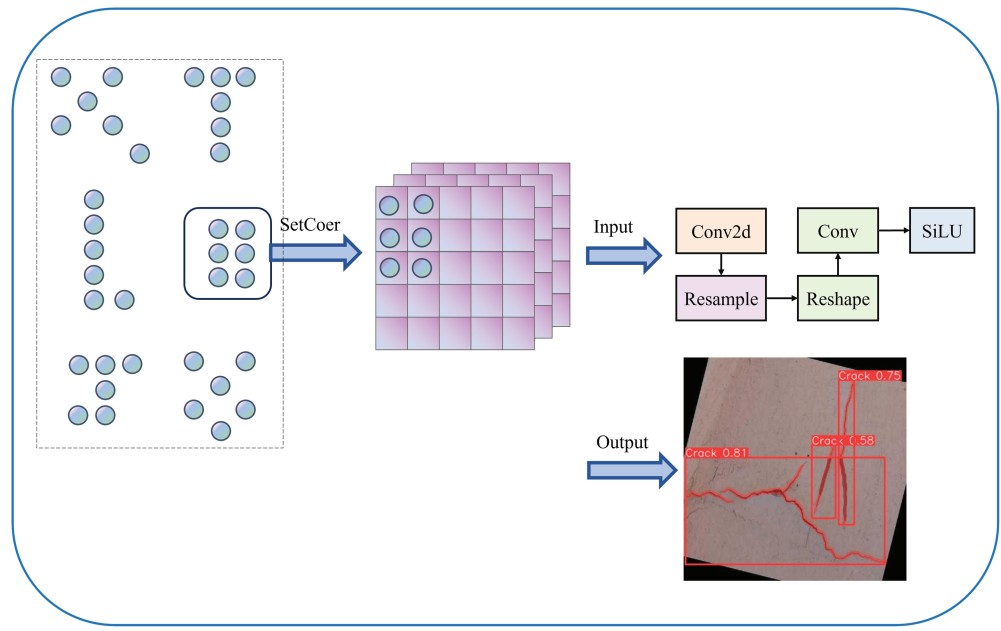

**Fig 7. Variable convolution process.**

and focus more on extracting target features, making it more suitable for the densely distributed one-dimensional nature of crack detection. Considering the inherently variable geometric characteristics of road cracks, and to better extract the features of cracks. We adopt a

staged feature extraction strategy in the backbone network part, first using two ordinary convolutions for the initial extraction of features and then combining DCNv-C2f and AKConv to optimize the ability of the model to extract features and reduce the computational complexity of the model by taking advantage of the variable convolution's substantial variability of feature extraction and high generalization Through this strategy ability, the backbone network can extract rich feature information every time to improve the detection accuracy and robustness of the model.

**Multiscale enhanced neck.** The neck network processes features extracted by the backbone network perform multi-scale fusion and feature enhancement and passes the processed features to the detection head for bounding box prediction and class classification. To better extract and process features from the backbone network, we continue the PAN-FPN concept from YOLOv8, using a top-down path with FPN for multi-scale feature fusion and a bottom-up path with PAN to transfer low-level features to high-level features, further enriching the multi-scale features. Traditional networks often face interference in feature extraction due to background noise and varying target sizes in complex backgrounds, which affects the model's ability to extract features. To enhance the model's capability to capture key features, we introduce the Global Attention Mechanism (GAM) in the neck network. This mechanism integrates global information, strengthening the relationships between different channels, thereby improving the model's adaptability to features at different scales. To enhance feature fusion efficiency, boost feature representation capability, and improve computational efficiency, we replace the standard convolution operations in YOLOv8 with AKConv operations in the neck network. We retain some C2f modules to optimize the flow of information, enabling efficient fusion of features across different levels and obtaining richer gradient flow information.

In the neck network, the attention mechanism plays a crucial role, allowing the model to focus on important regions in the image and assign appropriate weights, thereby enhancing the model's ability to locate and identify significant features. Traditional models typically apply the attention mechanism in a single dimension (either channel or spatial), exemplified by channel attention mechanisms like SENet [63] and spatial attention mechanisms like CBAM [64]. Although CBAM adds spatial information on top of SENet, it does not consider the importance of cross-dimensional interactions between channels and space, leading to the loss of information across dimensions. To address this issue, we propose the Global Attention Mechanism (GAM), which enhances global cross-dimensional interactions by amplifying global information exchange. By integrating spatial, channel, and temporal scale information, GAM significantly improves the accuracy and interpretability of detection results.

As shown in Fig 8, GAM captures global information through excellent global perception abilities, enhancing target processing capabilities. The main idea is to associate global contextual information with local features. It measures the similarity between global and local features at each position to emphasize local features corresponding to global features. This weighted feature fusion mechanism enables the model to better adapt to different tasks and scenarios. In the channel attention submodule, a 3D arrangement is used to retain information across all three dimensions. Two layers of Multi-Layer Perceptron (MLP) are employed to amplify cross-dimensional channel-space dependencies [65]. The spatial attention submodule focuses on spatial information by utilizing two convolutional layers for spatial information fusion. The entire process is illustrated in the accompanying figure and represented by the following equations.

$$\begin{cases} F_2 = M_c(F_1) \otimes F_1 \\ F_3 = M_s(F_2) \otimes F_2 \end{cases} \tag{6}$$

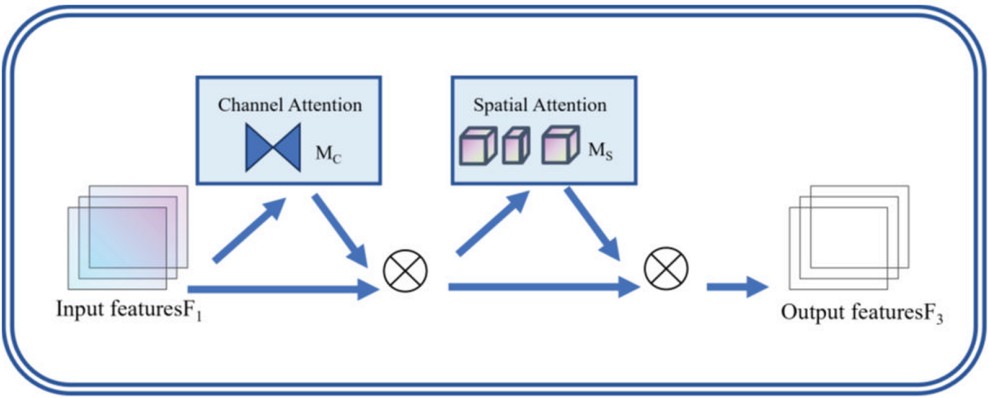

**Fig 8. GAM structure diagram.**

where: $F_1$ represents the input features, $C$ denotes the number of channels, $H$ indicates the height, and $W$ represents the width. $F_2$ signifies the intermediate state, and $F_3$ represents the output features. $M_c$ stands for the channel attention map, while $M_s$ denotes the spatial attention map. $\otimes$ represents element-wise multiplication [40].

The channel attention submodule and the spatial attention submodule are shown in Fig 9. In the channel attention submodule , the features are first subjected to initial channel transposition and then processed through a dual-layer multi-layer perceptron (MLP) to enhance their spatial and channel dimensions correlations.

The spatial attention submodule employs two (7x7) convolutional layers to capture detailed information in the spatial dimension.

The Global Attention Mechanism (GAM) operates on the head of the neck network. The last layer of the backbone network, the SPPF module, transforms features of different scales into fixed-size feature vectors. Once the GAM attention mechanism receives these feature

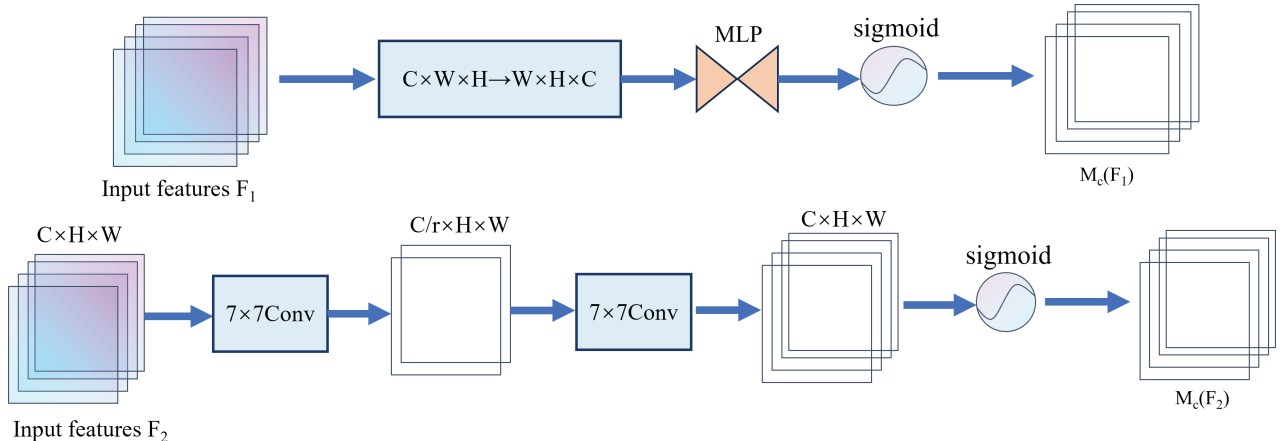

**Fig 9. Structure diagram of the attention mechanism submodule.**

vectors, they effectively retain the crack information extracted by the backbone network and pass it to the neck network for subsequent feature fusion, enhancing the fusion capabilities of the neck network. When facing complex background environments (such as occlusions and small target objects), this design significantly improves the model's ability to capture crack features, enhancing detection performance in multi-scale scenarios and addressing the challenges posed by background conditions in road crack detection. Given the multiple interferences in the data and the inherently variable geometric characteristics of cracks, the Global Attention Mechanism (GAM) emerges as the optimal choice for suppressing unimportant information in images while extracting crucial details.

**Lightweight detection head.** The head network is used to detect target categories and locations. YOLOv8 adopts a decoupled head architecture, which internally employs traditional convolutional computation methods. This approach generates a significant amount of redundant features, which significantly increase the parameters and computational load of the detection head, consume large amounts of device memory, and slow down the real-time performance of crack detection. All three detection heads employ a dual-branch structure, separating the tasks of category classification and bounding box position prediction. This strategy enables feature extraction across different scales and allows for more targeted task completion. In multi-task detection, it enhances detection robustness. However, this strategy significantly increases the model's computational and parameter demands in single-class tasks. Therefore, we designed a novel coupled head structure called the G-head (Ghost-Head), as shown in Fig 10. It uses a single-branch structure, sharing parameters for bounding box prediction and category classification, effectively reducing the computational burden. The coupled head minimizes the differences between various operations, making the model more robust to different inputs and improving its overall stability.

The GhostConv module from GhostNet [54] was incorporated into the detection head, as illustrated in Fig 11. It uses a small number of traditional convolutions to extract a limited set of features, then applies linear transformations (a computationally inexpensive operation) to generate more features. These are then identity-mapped to the original convolutional features, and the two sets are concatenated, achieving effects similar to traditional convolution but with a significant reduction in model parameters and computation. The G-head employs weight-sharing, effectively eliminating unnecessary convolutions. Feature extraction is performed through two GhostConv layers and a Conv2d operation, leveraging the efficient computational performance of GhostConv for detection tasks. This structure substantially reduces redundant features brought by traditional convolutions without losing critical information. It significantly reduces the model size, making it more lightweight. By filtering out excessive redundant features, the model minimizes interference from irrelevant information during detection and significantly improves its crack detection accuracy.

The loss function is a crucial numerical evaluation metric that measures the discrepancy between model predictions and actual outcomes. The CIoU regression loss employed by YOLOv8 considers three aspects: the overlap area of the bounding boxes, the distance between the center points, and the consistency of aspect ratios. However, it still has limitations when dealing with datasets that have quality issues. Therefore, we have replaced CIoU with Wise-IoU [40,55,66]. Using Wise-IoU effectively reduces the impact of low-quality samples on the overall evaluation. When the anchor boxes overlap with the target boxes, the influence of geometric factors on penalties diminishes, thereby enhancing the model's generalization ability.

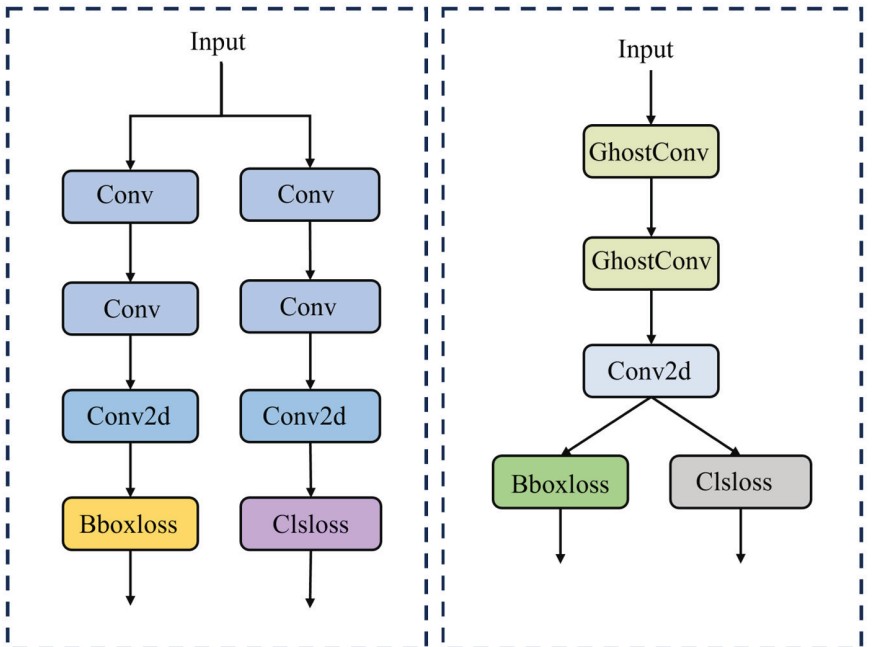

**Fig 10. Ghost detect module.**

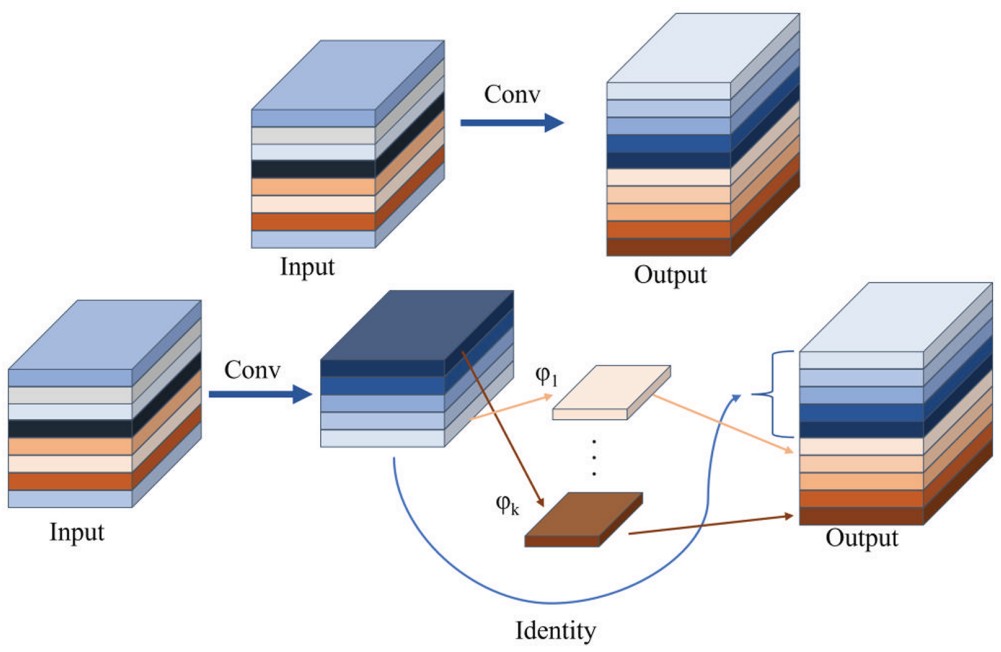

**Fig 11. Principle of ghost convolution.**

$$\begin{cases} L_{\text{WIoUv1}} = R_{\text{WIoU}} L_{\text{IoU}} \\ R_{\text{WIoU}} = \exp\left( \dfrac{(x-x_{gt})^2 + (y-y_{gt})^2}{(W_g^2 + H_g^2)^*} \right) \end{cases} \tag{7}$$

$$\begin{cases} \beta = \frac{L_{IoU}^{*}}{L_{IoU}^{**}} \in [0, +\infty) \\ L_{WIoUv3} = rL_{WIoUv1}, \quad r = \frac{\beta}{\delta\alpha^{\beta-\theta}} \end{cases} \quad (8)$$

In the formula: $R_{\mathrm{WIoU}}$ represents the penalty term,$L_{\mathrm{IoU}}$is the bounding box loss,$(x,y)$represents the location of the predicted box, $(x_{gt}, y_{gt})$ is the location of the ground truth box,$W_g, H_g$are the width and height, respectively,$\beta$ indicates outlier degree, $r$is the gradient gain, $\alpha,\theta$ are two hyperparameters,$\alpha$ adjusts the sensitivity of the loss function to the positions of the target box and predicted box, and modulates the penalty level for aspect ratio and bounding box scaling [67]. The G-Head module optimizes the bounding box prediction and category classification processes by integrating them into a unified workflow. This reduces the impact caused by computational divergence in dual-branch structures and minimizes unnecessary resource consumption. On this foundation, Wise-IoU effectively processes anchor boxes, significantly reducing the adverse impact of low-quality samples on model training. This strategy substantially enhances robustness and detection speed in complex environments, excelling particularly in real-time crack detection tasks.

## Experiment

### Experimental environment

The experimental setups used in this study, including the hardware platform, software environment, and hyperparameters, are detailed in Tables 1 and 2.

Except for those specifically modified, the hyperparameters used in this experiment remain consistent with the official source code. This setup ensures the consistency and comparability of the experiments, thereby enhancing the credibility of the experimental results and providing a solid foundation for model evaluation.

### Data set

The dataset used in this study comes from the Roboflow crack dataset [https://universe.roboflow.com/university-bswxt/crack-bphdr] and images of road cracks collected in the field.

**Table 1. Experimental environment.**

| Software or hardware platform | Model Version |
|---|---|
| CPU | Intel(R) Xeon(R) Gold 6342 CPU 2.80GHz |
| GPU | NVIDIA GeForce RTX 4090 24G |
| Operating System | Linux |
| Deep Learning Framework | PyTorch1.12.1 |
| Programming Language | Python3.8 |

**Table 2. Hyperparameter settings.**

| Parameter | Setting |
|---|---|
| Learning Rate | 0.01 |
| Image Size | 416x416 |
| Momentum | 0.937 |
| Optimizer | SGD |
| Batch Size | 64 |
| Epoch | 175 |

It consists of 4,040 road crack images annotated in YOLO format, all sized at 416 × 416 pixels. The dataset used in this study comes from the Roboflow crack dataset, which consists of field-collected images of road cracks. It includes 4,040 labeled images of road cracks in YOLO format, all sized at 416 × 416 pixels. The dataset encompasses various road types, including asphalt, concrete, and dirt roads, covering urban, town, rural, and forest areas. The dataset underwent multiple preprocessing techniques for enhancement using the image augmentation methods provided by the Roboflow platform to improve the model's generalization ability. Techniques applied include random angle rotation, cropping, brightness adjustments, and exposure adjustments. To ensure a reasonable data distribution and the reliability of test results, the dataset is divided into three parts: training set, validation set, and test set. Specifically, the training set contains 3,232 images, the validation set comprises 404 images, and the test set includes 404 images. This division guarantees diversity and sufficiency in training while providing ample validation and test data for model tuning and performance evaluation.

Sample images of the dataset are shown in Fig 12.

## Evaluation indicators

To test the detection performance of the improved model we proposed, we use Precision, Recall, F1-Score, and mAP as evaluation metrics. Precision refers to the ratio of samples predicted as positive to the actual positive samples; Recall indicates the proportion of correctly predicted positive samples out of all actual positive samples; F1-Score is the harmonic mean of Precision and Recall; mAP represents the mean average precision across all classes. The expressions for the four metrics are as follows:

$$\text{Precision} = \frac{TP}{TP + FP} \tag{9}$$

$$\text{Recall} = \frac{TP}{TP + FN} \tag{10}$$

$$F_1 = \frac{2 \cdot \text{Precision} \cdot \text{Recall}}{\text{Precision} + \text{Recall}} \tag{11}$$

$$mAP = \frac{\sum_{i=1}^{N} AP_i}{N} \tag{12}$$

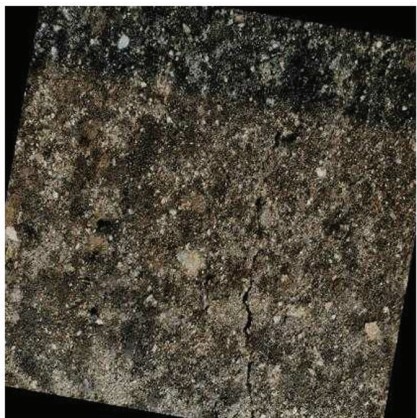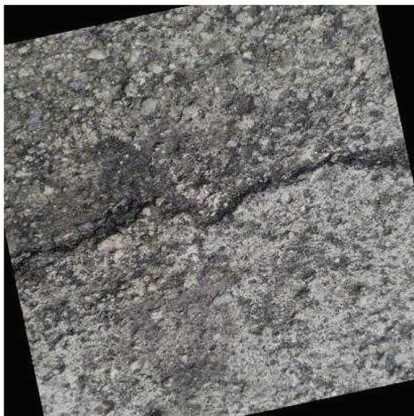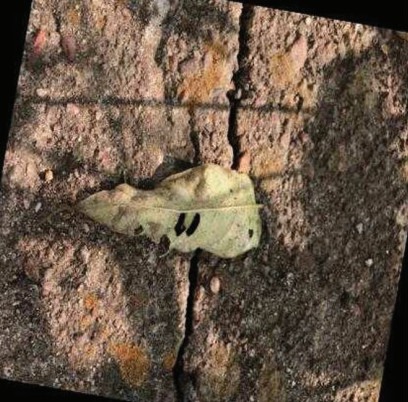

**Fig 12. Example of preprocessed image.**

where: *TP*: True Positive, predicted as a positive sample and actually is a positive sample; *FP*: False Positive, predicted as a positive sample but actually is a negative sample; *FN*: False Negative, predicted as a negative sample but actually is a positive sample; *AP*: Average Precision.

## Experimental results

**Ablation experiment.** The road crack detection model designed in this paper is based on YOLOv8, with improvements made to its backbone network, neck network, and detection head. The improved model has produced commendable results in terms of overall average precision after training. On the validation set, the model achieved 0.907 Precision, 0.782 Recall, 0.861 mAP@0.5, 0.653 mAP@0.5–0.95, and 7.6 GFLOPS. In comparison, the YOLOv8n model recorded 0.88 Precision, 0.735 Recall, 0.808 mAP@0.5, 0.614 mAP@0.5–0.95, and 8.1 GFLOPS. The proposed model demonstrates a 5.2% improvement in accuracy, a 4% increase in recall, a 1.5% rise in mAP@0.5, a 0.7% enhancement in mAP@0.5–0.95, and a reduction of 0.4 in GFLOPS. The results indicate that the YOLO (Improved) model proposed in this paper significantly enhances crack detection effectiveness while being more lightweight.

To systematically evaluate the performance gaps between each module, this study designed the following comparisons, utilizing Precision, Recall, mean Average Precision (mAP), computational complexity (GFLOPS), and F1 score as evaluation metrics, as detailed in Table 3.

The experimental results indicate that replacing the CIoU loss function with Wise-IoU in YOLOv8n improved the model's accuracy by 2.3% and mAP@0.5 by 0.6%, with no computational load increase and a slight enhancement in model performance. This improvement validates Wise-IoU's effectiveness in mitigating interference from low-quality sample annotations and handling samples of varying quality. The DCNv-C2F module, compared to YOLOv8n, reduced GFLOPS by 0.2, maintained accuracy similar to the baseline model, and showed significant improvements in other metrics. This suggests that the module effectively reduces information redundancy in feature maps and promotes information sharing between convolutional layers while enhancing elongated features' extraction capability. The introduction of the AKConv module resulted in a 3.6% increase in model accuracy and a 1.6% increase in mAP@0.5 compared to YOLOv8n. This improvement affirms the effectiveness of deformable convolution kernels in feature extraction and fusion, further enhancing the model's adaptability and ability to represent multi-scale features. The experimental results of the GAM module show that although there was a slight decrease in model accuracy, other metrics improved. This demonstrates the significant advantage of the GAM module in global information extraction, allowing for better capture of feature details and leading the model to detect more target objects overall.

By integrating the aforementioned improvements and incorporating the proposed detection head module, the Flexi-YOLO model demonstrates a 2.7% increase in accuracy, a 4.7% increase in recall, a 5.3-point enhancement in mAP@0.5, and a 3.9-point improvement in mAP@0.5–0.95 compared to the baseline model. GFLOPS decreased by 0.5, while the F1 score improved from 0.80 to 0.84. These significant increases validate the superiority of Flexi-YOLO in crack detection tasks, leading to a comprehensive enhancement of its detection capabilities. The reduction in GFLOPS indicates that the proposed detection head effectively filters redundant information, extracts valuable features, and achieves model lightweight. These enhancements effectively prevent the loss of contextual information during the multi-level fusion process, facilitate information integration across different channels, and significantly improve the overall performance metrics of the model.

**Table 3. Ablation experiment.**

| Model | DCNv-C2f | AKConv | GAM | Wise-IoU | G -Head | Precision | Recall | mAP@0.5 | mAP@0.5-0.95 | GFLOPS | F1 |
|-------|----------|--------|-----|----------|---------|-----------|--------|---------|--------------|--------|-----|
| A |  |  |  |  |  | 0.88 | 0.735 | 0.808 | 0.614 | 8.1 | 0.801 |
| B |  |  |  | √ |  | 0.903 | 0.735 | 0.814 | 0.616 | 8.1 | 0.810 |
| C | √ |  |  |  |  | 0.89 | 0.744 | 0.825 | 0.64 | 7.9 | 0.64 |
| D |  | √ |  |  |  | 0.916 | 0.711 | 0.824 | 0.604 | 8.0 | 0.604 |
| E |  |  | √ |  |  | 0.886 | 0.751 | 0.827 | 0.61 | 9.4 | 0.61 |
| F | √ | √ | √ | √ |  | 0.917 | 0.751 | 0.831 | 0.64 | 9.3 | 0.64 |
| G | √ | √ | √ | √ | √ | 0.907 | 0.782 | 0.861 | 0.653 | 7.6 | 0.84 |

To validate the improvements in our model, we will visually compare the enhanced model with YOLOv8n.

The changes in the mAP value are shown in Fig 13. Compared to YOLOv8, our model has smaller fluctuations during the training process and demonstrates better convergence. In the early stages of numerical training, it converges more quickly, with values consistently higher than those of YOLOv8, confirming that Flexi-YOLO exhibits stronger performance and greater robustness.

The loss change curve is shown in Fig 14. From the loss curve, it can be observed that the loss of Flexi-YOLO after convergence is lower than that of YOLOv8, and it converges more quickly. This indicates that our model performs better and shows superior results in detection tasks.

The PR curve is shown in Fig 15, indicating that Flexi-YOLO outperforms YOLOv8n with a higher mAP@0.5, increasing from 80.8% to 86.1%. Additionally, the curve is smoother, suggesting that our model possesses greater robustness and stability. The area under our model's curve exceeds that of the YOLOv8 curve, demonstrating superior performance in road crack detection tasks.

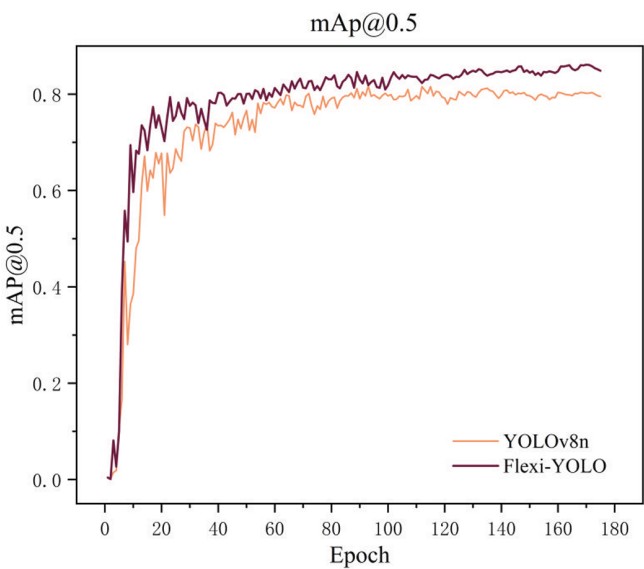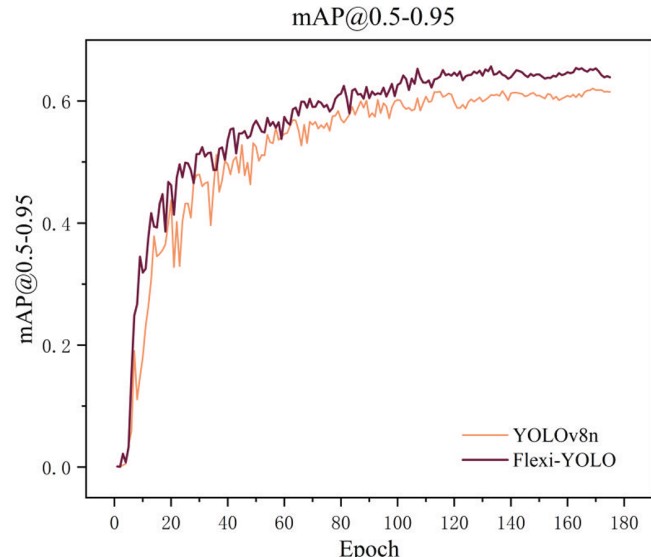

**Fig 13. mAP comparison.**

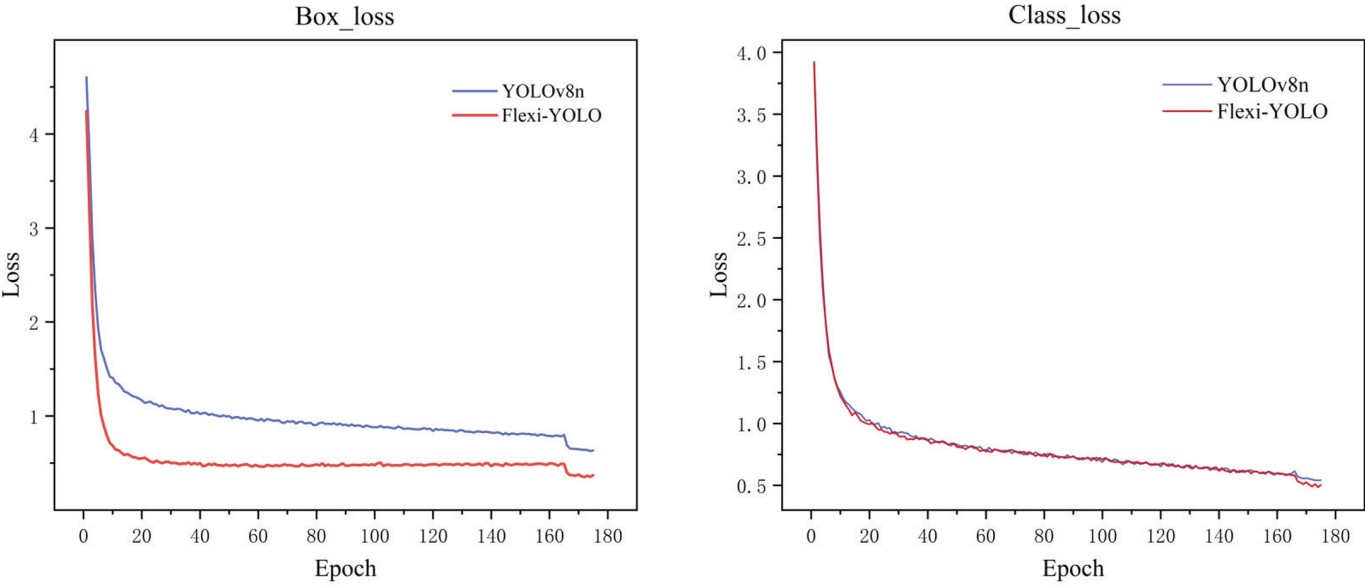

**Fig 14. Loss curve.**

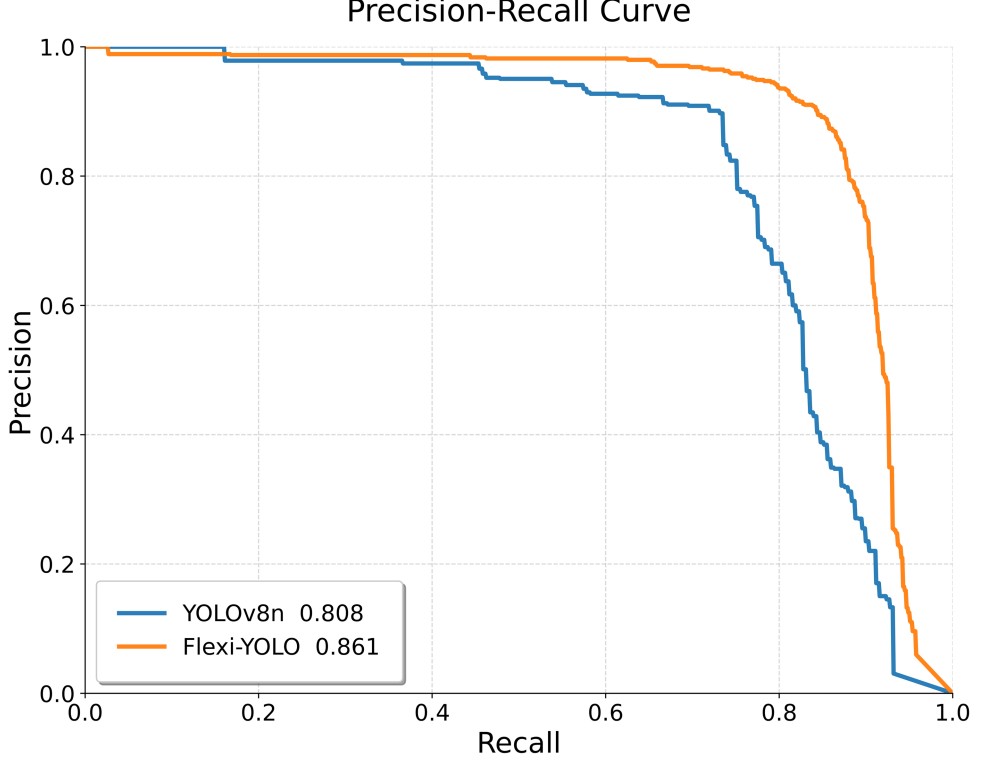

**Fig 15. PR comparison.**

**Comparative experiments.** To further demonstrate the superiority of the method proposed in this paper, we compared the model with YOLOv5, YOLOv7, YOLOv8, YOLO9, YOLOv10,and RT-DETR, as shown in Table 4.

**Table 4. Comparative experiments.**

| Model | Precision | Recall | mAP@0.5 | mAP@0.5-0.95 | GFLOPS | F1 |
|---|---|---|---|---|---|---|
| YOLOv5n | 0.841 | 0.727 | 0.799 | 0.546 | 4.1 | 0.779 |
| YOLOv7tiny | 0.826 | 0.687 | 0.737 | 0.511 | 13.0 | 0.750 |
| YOLOv8n | 0.88 | 0.735 | 0.808 | 0.614 | 8.1 | 0.808 |
| YOLOv9-t | 0.88 | 0.73 | 0.818 | 0.63 | 10.7 | 0.798 |
| YOLOv10n | 0.842 | 0.728 | 0.806 | 0.608 | 8.2 | 0.780 |
| RT-DETR | 0.915 | 0.736 | 0.831 | 0.653 | 56.9 | 0.81 |
| Ours | 0.907 | 0.782 | 0.861 | 0.653 | 7.6 | 0.84 |

Based on the data in the table, the improved model outperforms others across four or five evaluation metrics: Precision, Recall, mAP@0.5, mAP@0.5–0.95, and F1 score, demonstrating significantly superior detection performance. The experimental results indicate that our proposed model significantly enhances detection capabilities without consuming excessive resources.

To visually demonstrate the improved effects of our model, we conducted inference experiments, comparing it with YOLOv5n, YOLOv7tiny, YOLOv8n, YOLOv9t, YOLOv10n, and RT-DETR. We selected a diverse set of cracks with varying scales and shapes as our experimental data, ensuring a large quantity and variety.

The experimental results are shown in Fig 16, where a represents YOLOv5n, b represents YOLOv7tiny, c represents YOLOv8n, d represents YOLOv9t, e represents YOLOv10n, f represents RT-DETR, and g represents Flexi-YOLO. The results indicate that the proposed Flexi-YOLO model significantly outperforms other comparative models regarding detection accuracy and localization precision. It is noteworthy that although other models can detect most cracks, they still exhibit various shortcomings in practical applications. For instance, YOLOv5n and YOLOv9t display lower detection accuracy and a high rate of missed detections. Moreover, YOLOv5n, YOLOv8n, YOLOv10n, and RT-DETR demonstrate significant issues with overlapping detection boxes during detection. All comparative models fail to accurately detect targets in complex scenarios such as shadows, low lighting, low contrast, and small cracks, resulting in high missed detection rates. Additionally, these models often redundantly detect inevitable cracks when generating prediction boxes. The high similarity between background noise and crack features leads to frequent false detections. Specifically, the Transformer-based RT-DETR, while slightly superior in detection accuracy compared to other models, suffers from even more severe false and missed detection issues. Its substantial computational load of up to 56.9 GFLOPS places it at a disadvantage in computational efficiency, significantly diminishing its cost-effectiveness.

In contrast, the Flexi-YOLO model proposed in this paper accurately detects all types of cracks and exhibits higher detection accuracy, more precise edge localization, lower miss rates, and superior computational efficiency. These advantages enable Flexi-YOLO to demonstrate greater adaptability and performance in detection tasks within complex environments. A visual analysis using Gradient-weighted Class Activation Mapping (Grad-CAM) [68] on the SPPF layer of YOLOv8n and the GAM attention mechanism layer of the Flexi-YOLO model enhances the model's interpretability and effectiveness. Grad-CAM calculates gradients based on class confidence scores and generates corresponding weights. This approach allows for an intuitive understanding of the feature representation among different modules and network layers, particularly regarding the focus of features in the complex task of detecting road cracks in cluttered backgrounds when using Flexi-YOLO.

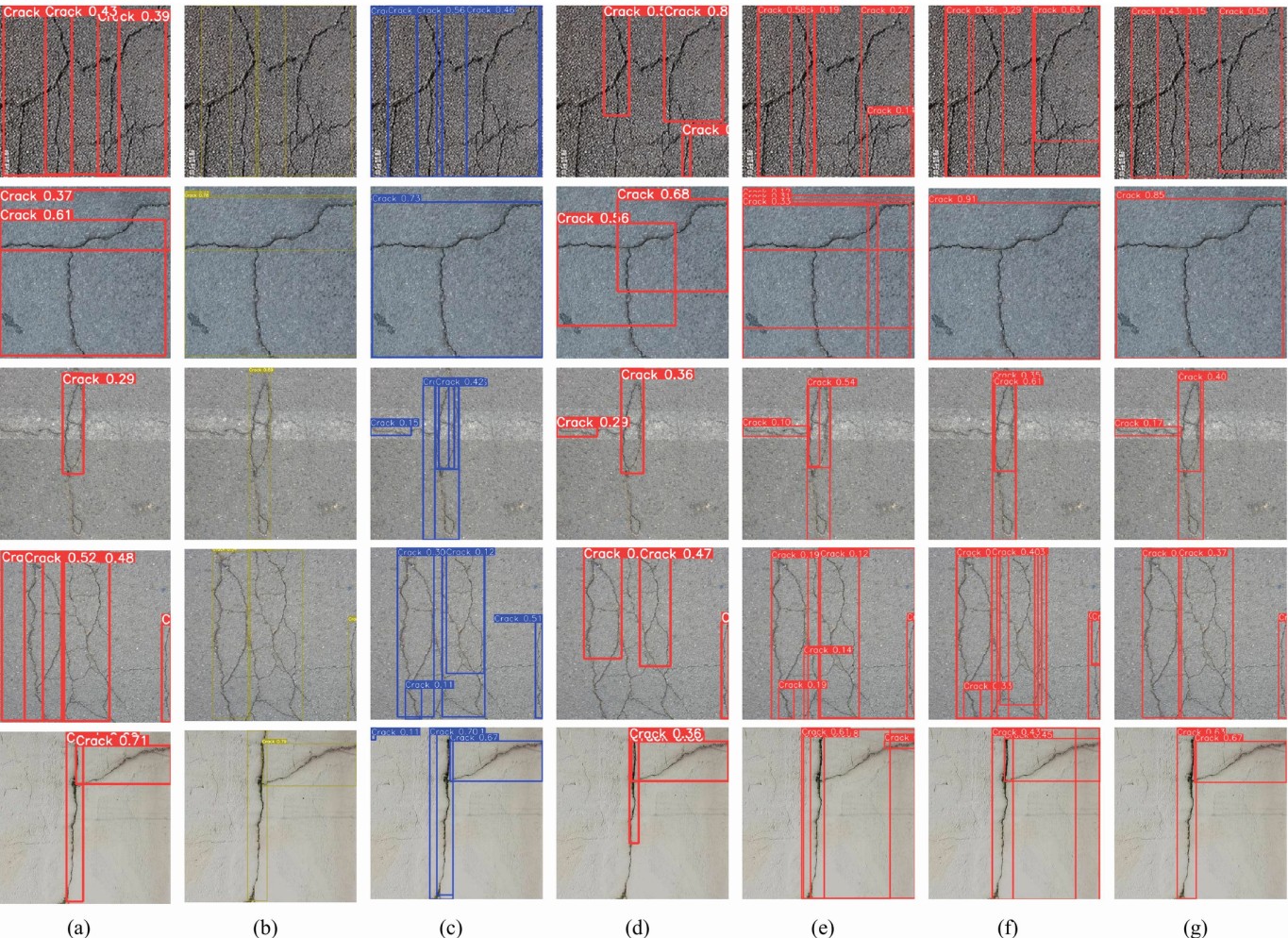

**Fig 16. Results of the inference experiment.**

The experimental results are shown in Fig 17. The SPPF layer of the backbone network of the baseline model aggregates multi-scale contextual features through spatial pyramid pooling, and its fixed receptive field restricts the adaptability to irregular crack patterns. In contrast, Flexi-YOLO focuses on feature details through its convolution operation with adaptive sampling points and the GAM attention mechanism, enabling the model to focus more on the target region.Specifically, when the original image presents cracks in streaks or grids, the baseline model can only focus on the central parts of the cracks and fails to address the edges. Additionally, when cracks bifurcate, the baseline model only recognizes parts of one crack while neglecting the other bifurcated portion. On the other hand, Flexi-YOLO benefits from its excellent adaptive sampling mechanism, which dynamically adjusts the positions of sampling points based on the shape of the cracks. In contrast, the GAM attention mechanism retains important features and suppresses ineffective ones. By visualizing the model's heat maps, the experimental results further illustrate the reasons for and effectiveness of Flexi-YOLO in avoiding missed detections and false positives.

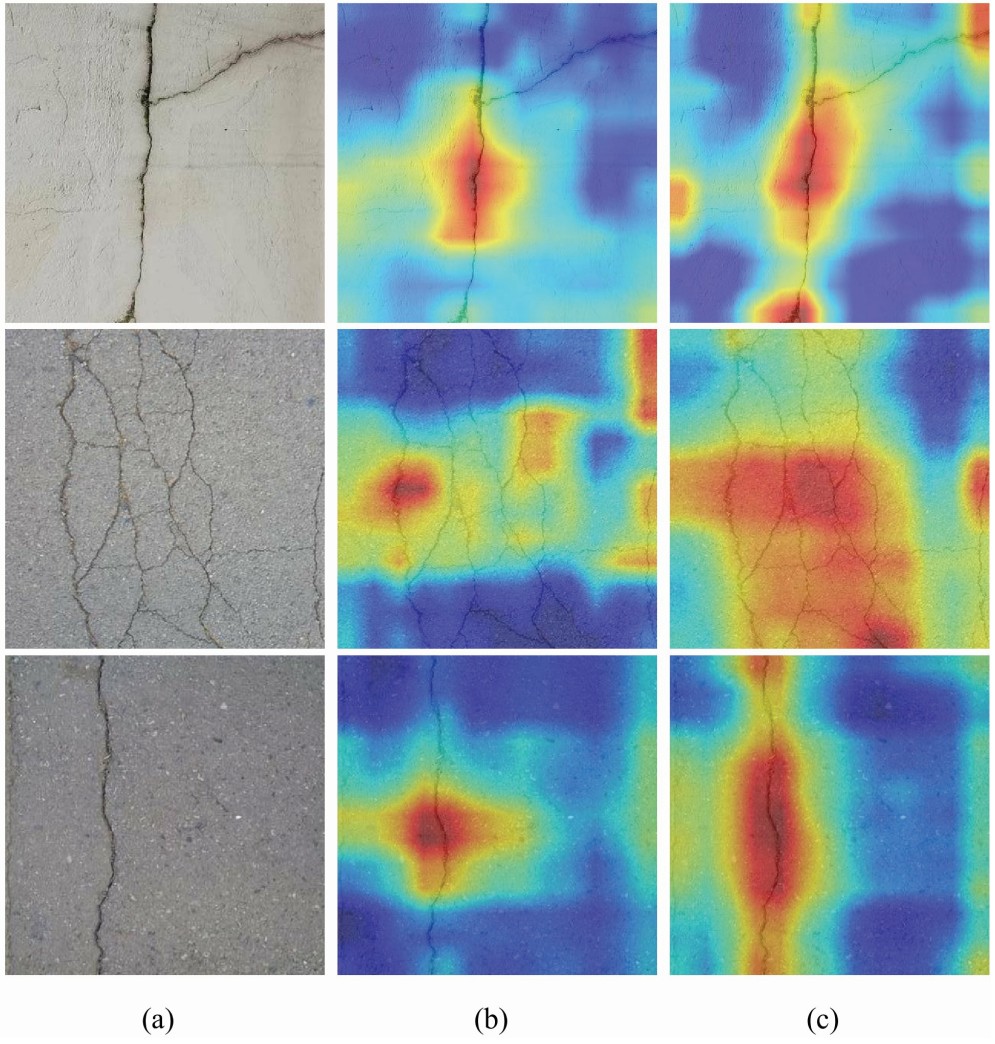

(a) (b) (c)

**Fig 17. Thermodynamic chart.**

**Model generalization experiment.** In order to comprehensively assess the generalization performance of the Flexi-YOLO model across different datasets, we designed a multidimensional comparative experimental framework. The data sources for the experiment are divided into two parts: 1. The self-constructed DATA6000 dataset, consisting of 6,000 images that cover various road scenes (such as urban roads, highways, and rural roads), different lighting conditions (sunny, cloudy, and nighttime), and multiple types of cracks (transverse cracks, longitudinal cracks, and network cracks), sourced from the integrated datasets of the Roboflow website. 2. The RDD2022 dataset [69], which includes four categories of typical road damage: longitudinal cracks (D00), transverse cracks (D10), network cracks (D20), and potholes (D40).

In the experimental design and dataset processing, we adopted a dual-control strategy: Horizontal control involves introducing the RDD2022 dataset to partition it into subsets from different countries and regions, validating the model's adaptability in various areas and environments. Vertical control divides the dataset according to the specific characteristics of

road detection scenarios, distinguishing between ground view and aerial view. This further subdivides the RDD2022 dataset into two categories: low-angle motorcycle views (MotorBike) and drone aerial views to address differences in equipment perspectives. Flexi-YOLO model was specially optimized for the geometric features of linear cracks, its parameter update process was not explicitly tailored to network cracks (D20) and potholes (D40). However, the excellent generalization ability of Flexi-YOLO ensures that optimizing D00 and D10 categories does not negatively impact other damage types. Consequently, the other damage types have been excluded from the comparative experiments in this study. Ultimately, three sub-datasets were formed to conduct comparative experiments in different scenarios.

The Data6000 dataset consists of 6,000 images, while the MotorBike dataset contains 1,934 images, and the Drone dataset has 1,917 images. The internal annotation distribution of these different datasets is shown in Table 5. The annotation quantities for the D00 and D10 categories are relatively balanced, which aids the model's comprehensive feature learning, enhancing its generalization capability and accuracy in recognition. This helps effectively mitigate various negative impacts caused by insufficient data. A sufficient amount of labeled data provides a reliable foundation for feature extraction and parameter optimization during the model training process, enabling Flexi-YOLO to maintain stable detection performance during evaluations.

For data partitioning, we divided all datasets into a training set, validation set, and test set in an 8:1:1 ratio to ensure that the model could adequately learn the features during training while preventing the risk of overfitting.

We selected YOLOv8n as the baseline model and conducted a comprehensive evaluation based on four core metrics: Precision, Recall, mean Average Precision at IoU=0.5 (mAP@0.5), and mean Average Precision at IoU from 0.5 to 0.95 (mAP@0.5–0.95).

The experimental results are shown in Table 6, where the Flexi-YOLO model demonstrates significant advantages across three datasets. On the DATA6000 dataset, Recall improved by 6%, andmAP@0.5–0.95 increased by 4.6%. In the MotorBrike subset, all detection accuracy metrics improved: for classes D00/D10, Precision (P) increased by 4.5%/5.6%, Recall (R) enhanced by 2%/3.2%, mAP@0.5 rose by 3.6%/2.8%, and mAP@0.5–0.95 grew by 4.4%/4.5%. In the drone dataset, Recall for classes D00/D10 improved by 9%/4.7%, mAP@0.5 increased by 2%/1.9%, and mA@0.5–0.95 saw enhancements of 2.2%/0.1%. These experimental results strongly confirm that Flexi-YOLO significantly enhances the generalization capability for multi-scenario road damage detection through architectural improvements and optimized training strategies, providing a reliable technical foundation and approach for future engineering deployment.

**Computational analysis of low-power devices.** To validate the feasibility of the Flexi-YOLO model proposed in this paper for practical applications, we deployed and tested the model on an NVIDIA GeForce RTX 1650 graphics card. The RTX 1650 is a cost-effective entry-level GPU with 4GB of memory and 896 CUDA cores, providing a computational power of approximately 2.9 TFLOPS. Its limited performance is suitable for simulating devices

**Table 5. Distribution of dataset annotations.**

| Dataset | Category | Annotation Count |
|---|---|---|
| Data6000 | Crack | 9120 |
| MotorBrike | D00 | 2678 |
| | D10 | 1096 |
| Drone | D00 | 1421 |
| | D10 | 1263 |

**Table 6. Model generalization experiment.**

| Model | Label | Precision | Recall | mAP@0.5 | mAP@0.5–0.95 |
|---|---|---|---|---|---|
| DATA6000-v8n | Crack | 0.76 | 0.545 | 0.633 | 0.409 |
| DATA6000-Flexi | Crack | 0.76 | 0.605 | 0.669 | 0.455 |
| Motor-v8n | D00 | 0.815 | 0.815 | 0.853 | 0.52 |
| | D10 | 0.809 | 0.764 | 0.845 | 0.459 |
| Motor-Flexi | D00 | 0.86 | 0.835 | 0.889 | 0.564 |
| | D10 | 0.865 | 0.796 | 0.873 | 0.504 |
| Drone-v8n | D00 | 0.738 | 0.647 | 0.722 | 0.356 |
| | D10 | 0.799 | 0.684 | 0.733 | 0.380 |
| Drone-Flexi | D00 | 0.728 | 0.737 | 0.742 | 0.378 |
| | D10 | 0.761 | 0.731 | 0.742 | 0.379 |

with similarly constrained computing capabilities and evaluating our model's performance. To assess the model's robustness, stability, and lightweight performance in various real-world environments, we captured images of road cracks in different settings as test data. In the images, 'a' represents the original image, 'b' corresponds to YOLOv8n, and 'c' denotes Flexi-YOLO.

The experimental results shown in Fig 18 indicate that the model exhibits outstanding inference capabilities on the RTX 1650 graphics card. In practical detection scenarios, the model demonstrates good robustness and detection accuracy. The baseline model, however, exhibited many false positives and missed detections in real-world environments, and its performance in locating cracks and forming detection anchor boxes was less than satisfactory. In contrast, Flexi-YOLO performed exceptionally well in practical detection environments. It could accurately identify the position and shape of road cracks even under challenging conditions, such as intense light, shadows, and mottled backgrounds. It successfully categorized multiple cracks in road images into distinct subsets, with a lower occurrence rate of false detections. Furthermore, its detection accuracy surpassed that of the baseline model. Regarding resource consumption, Flexi-YOLO occupies only 1135 MiB of memory, well below the entry-level memory of various mainstream mobile devices (4GB), achieving a real-time processing speed of 59 FPS, significantly exceeding the 30 FPS standard for real-time processing.

Validation through real-world detection confirms the effectiveness of the proposed improvements regarding lightweight design and practical application. The results demonstrate that the model can operate efficiently on resource-constrained devices through reasonable model design and lightweight optimization while maintaining high detection accuracy and real-time performance. This highlights the significant potential application of the proposed method, which meets the requirements for light weight and efficiency in practical engineering contexts.

## Conclusion

This article presents a novel Flexi-YOLO model applied to road crack detection. The model employs DCNv-C2f adaptive convolution sampling positions and AKConv adaptive convolution kernel sizes within its backbone network to enhance its ability to extract crack features, allowing for more accurate and flexible crack identification. By adding a GAM attention mechanism at the neck of the network, the model can comprehensively capture global information and further fuse detailed features with low-level network characteristics to obtain more feature information. The newly designed detection head mitigates the issue of information redundancy caused by the attention mechanism's extensive global information collection and the increased model size due to the high parameter count and computation in

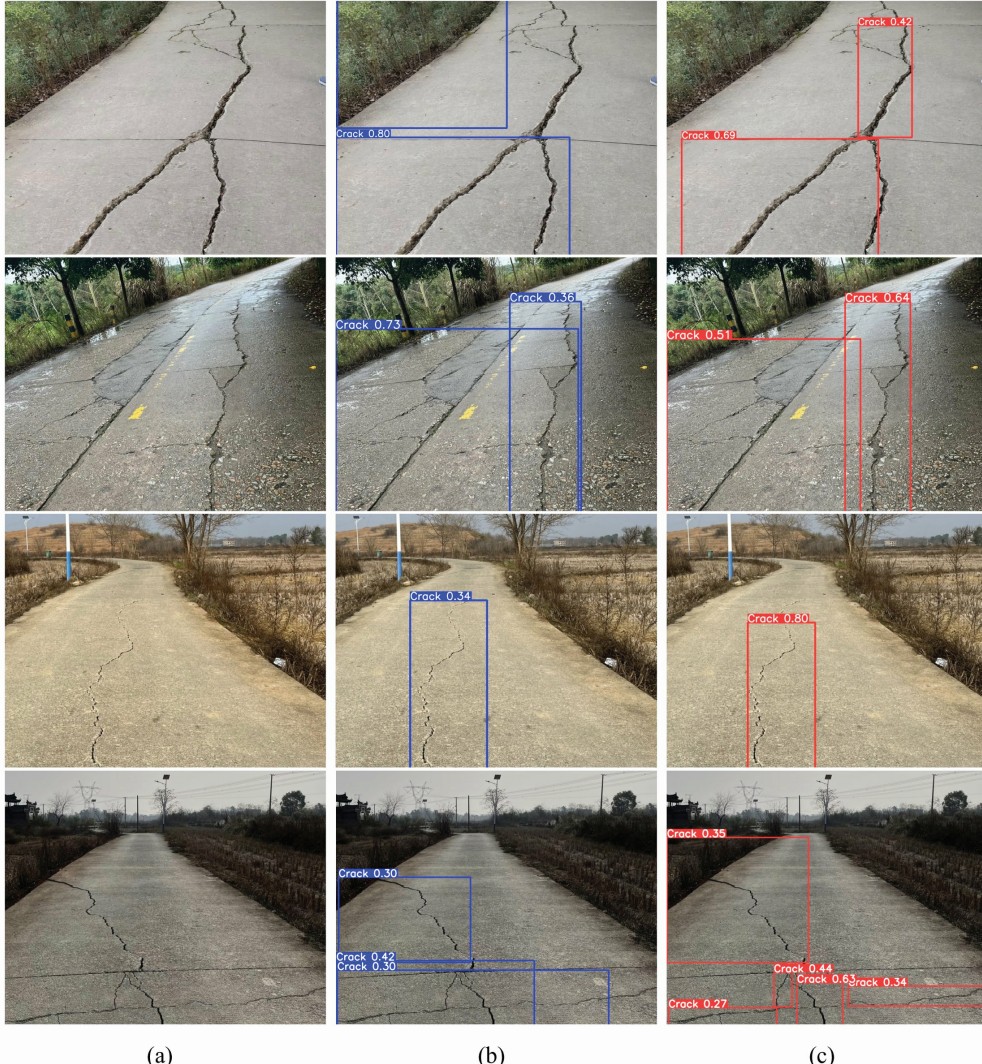

**Fig 18. Actual inspection image.**

the original detection head, making it challenging to deploy on devices with limited computational resources. The Wise-IoU method assigns different weights based on sample quality, effectively reducing the negative impact of low-quality samples on the model. With these improvements, the YOLOv8 model performs better in handling complex information across various road conditions, demonstrating stability against fine details and interference information. The developed model shows outstanding performance across various metrics, specifically a 2.7% increase in accuracy, a 4.7% increase in recall, a 5.3% improvement in mAP@0.5, a 3.9 improvement in mAP@0.5-0.95, a 0.5 decrease in GFLOPS, and an increase in F1 score from 0.80 to 0.84. This data underscores the superiority of Flexi-YOLO in multiple dimensions. Furthermore, practical detection experiments indicate that our model excels in numerical evaluation and real-world road condition detection. This validates the practicality of our model improvements and provides valuable experience and technological advancement for real-world road crack operations, offering reliable solutions for global road aging, safety,

and maintenance. Its lightweight design also presents an economically efficient solution for infrastructure maintenance, contributing to global public safety and sustainable development.

## Acknowledgments

We would like to express our gratitude to Professors Xiangli Yang,Lei Zhang,Zhangli Lan and Di Wang from Chongqing Jiaotong University for their thoughtful feedback on the manuscript and language editing. We appreciate the time and effort contributed by all research participants.

## Author contributions

**Conceptualization:** Jiexiang Yang, Renjie Tian, Zexing Zhou.

**Data curation:** Jiexiang Yang, Zexing Zhou, Xingyue Tan.

**Formal analysis:** Jiexiang Yang, Renjie Tian, Xingyue Tan.

**Investigation:** Renjie Tian, Pingyang He.

**Methodology:** Jiexiang Yang, Renjie Tian, Zexing Zhou.

**Project administration:** Jiexiang Yang, Renjie Tian.

**Resources:** Jiexiang Yang, Renjie Tian.

**Supervision:** Jiexiang Yang.

**Validation:** Jiexiang Yang, Zexing Zhou, Pingyang He.

**Visualization:** Jiexiang Yang, Renjie Tian, Zexing Zhou, Xingyue Tan, Pingyang He.

**Writing – original draft:** Jiexiang Yang, Renjie Tian, Zexing Zhou, Xingyue Tan, Pingyang He.

**Writing – review & editing:** Jiexiang Yang, Renjie Tian, Zexing Zhou, Xingyue Tan, Pingyang He.

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
