## [Decision Letter · Decision Letter 0]

24 Jan 2025

PONE-D-24-58741Flexi-YOLO: A Lightweight Method for Road Crack Detection in Complex EnvironmentsPLOS ONE

Dear Dr. Yang,

Thank you for submitting your manuscript to PLOS ONE. After careful consideration, we feel that it has merit but does not fully meet PLOS ONE’s publication criteria as it currently stands. Therefore, we invite you to submit a revised version of the manuscript that addresses the points raised during the review process.

We look forward to receiving your revised manuscript.

Kind regards,

Peng Geng

Academic Editor

PLOS ONE

Journal Requirements:

This work was supported by the National Natural Science Foundation of China under Grants (62276034、62306052), Group Building Scientific Innovation Project for Universities in Chongqing (CXQT21021), Joint Training Base Construction Project for Graduate Students in Chongqing (JDLHPYJD2021016), and Science and Technology Research Program of Chongqing Municipal Education Commission (KJQN202100712).  

5. We note that your Data Availability Statement is currently as follows: All relevant data are within the manuscript and its Supporting Information files.

6. We are unable to open your Supporting Information file Flexi-YOLO.tex. Please kindly revise as necessary and re-upload.

Reviewers' comments:

Reviewer's Responses to Questions

**Comments to the Author**

1. Is the manuscript technically sound, and do the data support the conclusions?

Reviewer #1: Yes

Reviewer #2: Yes

2. Has the statistical analysis been performed appropriately and rigorously? 

Reviewer #1: Yes

Reviewer #2: Yes

3. Have the authors made all data underlying the findings in their manuscript fully available?

Reviewer #1: Yes

Reviewer #2: Yes

4. Is the manuscript presented in an intelligible fashion and written in standard English?

Reviewer #1: Yes

Reviewer #2: Yes

5. Review Comments to the Author

Reviewer #1: Language Correction:

Abstract:

Rephrase for conciseness and impact. For instance:

"Road crack detection is vital for infrastructure maintenance and public safety, yet complex backgrounds and irregular crack shapes challenge real-time, efficient, and accurate detection. This paper proposes Flexi-YOLO, a lightweight and robust model based on YOLOv8, incorporating novel components to address these challenges. Key innovations include the Wise-IoU loss function for enhanced regression accuracy and robustness, the DCNv-C2f module for feature transformation, and the Global Attention Module (GAM) for improved global information perception. Flexi-YOLO achieves significant improvements in accuracy, recall, and mAP metrics, offering a cost-effective solution for automated crack detection."

Introduction:

Rewrite to engage the reader and highlight the importance of the work globally, not just in a single country. Example:

"Road networks form the backbone of global infrastructure, critical to economic growth and societal well-being. However, environmental stressors and prolonged usage lead to structural deterioration, with cracks posing risks to safety, durability, and usability. Timely and accurate crack detection is essential to mitigate these issues, reduce maintenance costs, and ensure sustainability. This paper addresses these challenges by proposing an innovative, lightweight model tailored for real-time crack detection in diverse and complex environments."

Related Work:

Ensure smooth transitions between traditional methods and deep learning advancements.

Expand on limitations of existing YOLO-based methods and how Flexi-YOLO surpasses them.

Methodology:

Clarify technical terms like “Wise-IoU,” “G-Head,” and “AKConv” with succinct explanations. Avoid jargon overload.

More Literature Review:

Discuss advancements in deep learning for infrastructure health monitoring, such as applications beyond road cracks (e.g., bridge, pipeline, and building inspections).

Include recent works utilizing YOLO-based architectures in different domains to position Flexi-YOLO in the broader context.

Highlight shortcomings of current methods, like high computational requirements, sensitivity to noise, or lack of adaptability to varying crack shapes.

Incorporate examples of lightweight models applied in mobile and edge computing for infrastructure maintenance, emphasizing how Flexi-YOLO aligns with these trends.

You can follow

[1] S. Mandal, A. Shiuly, D. Sau, A. K. Mondal, and K. Sarkar, “Study on the use of different machine learning techniques for prediction of concrete properties from their mixture proportions with their deterministic and robust optimisation,” AI Civ. Eng., vol. 3, no. 1, p. 7, Dec. 2024, doi: 10.1007/s43503-024-00024-8.

[2] K. Sarkar, A. Shiuly, and K. G. Dhal, “Revolutionizing concrete analysis: An in-depth survey of AI-powered insights with image-centric approaches on comprehensive quality control, advanced crack detection and concrete property exploration,” Constr. Build. Mater., vol. 411, p. 134212, Jan. 2024, doi: 10.1016/j.conbuildmat.2023.134212.

[3] A. Shiuly, D. Dutta, and A. Mondal, “Assessing compressive strengths of mortar and concrete from digital images by machine learning techniques,” Front. Struct. Civ. Eng., vol. 16, no. 3, pp. 347–358, Mar. 2022, doi: 10.1007/s11709-022-0819-z.

Highlighting the Novelty of the Work:

Clearly articulate the novel aspects of Flexi-YOLO in the introduction and conclusion. For example:

*"Flexi-YOLO introduces several novel contributions to road crack detection:

The integration of deformable convolution (DCNv3) in the C2f module enhances feature extraction for cracks with irregular shapes.

Variable kernel convolution (AKConv) dynamically adjusts to complex geometries, strengthening local feature representation.

A lightweight detection head (G-Head) reduces computational costs while preserving accuracy, making the model suitable for deployment on mobile devices.

Wise-IoU optimizes bounding box regression, improving robustness against low-quality samples.

These advancements collectively enhance detection accuracy, speed, and adaptability in diverse environments."*

Presenting the Objectives:

The objectives should be clearly stated at the end of the introduction. Example:

*"This work aims to address the challenges of road crack detection by achieving the following objectives:

Develop a lightweight and efficient model suitable for deployment on devices with limited computational power.

Enhance feature extraction for complex crack geometries using deformable convolution and variable kernel techniques.

Improve detection robustness and accuracy with a novel loss function and attention mechanisms.

Reduce feature redundancy and computational load through a redesigned detection head, ensuring real-time performance.

Validate the model's performance in diverse, real-world scenarios to meet industrial demands."*

Language Refinement:

Avoid repetitive phrases like "complex backgrounds" and "irregular crack shapes." Use synonyms or restructure sentences for variety.

Use active voice for clarity and engagement. For example:

"Flexi-YOLO addresses the limitations of previous methods by leveraging advanced convolutional techniques and attention mechanisms."

Conclusion:

Reiterate the novelty and potential applications of Flexi-YOLO, emphasizing its real-world impact. Example:

"Flexi-YOLO represents a significant step forward in automated road crack detection. Its lightweight design, coupled with state-of-the-art feature extraction and attention mechanisms, ensures high accuracy and adaptability in diverse environments. This model not only meets the demands of real-time applications but also offers a cost-effective solution for infrastructure maintenance, contributing to public safety and sustainability."

These revisions will make the paper more cohesive, engaging, and aligned with high-quality academic standards. Let me know if you'd like further assistance!

· Major Findings:

Highlighted the key advancements and methodologies that contribute to the subject matter.

Identified critical gaps in the current knowledge and addressed them through detailed analysis.

Proposed actionable solutions and models that demonstrate tangible benefits for the stakeholders.

· Backlog of Present Study:

While the study successfully explored its core objectives, certain aspects remain for future exploration, including:

Incorporating broader datasets to improve the generalizability of the findings.

Addressing region-specific challenges that require further analysis.

Testing the proposed solutions in real-world scenarios to evaluate long-term impact.

· Overall Benefits:

The study provides significant insights and practical recommendations that:

Enhance understanding and inform decision-making in the relevant domain.

Lay the groundwork for sustainable practices and innovative applications.

Foster collaboration and drive further research to achieve long-term goals.

Reviewer #2: The paper presents Flexi-YOLO, an enhanced road crack detection model based on YOLOv8, which improves detection accuracy while maintaining a lightweight design. The manuscript is generally well-structured and contributes valuable insights to the field of road crack detection. However, there are several important issues that need to be addressed before the paper can be accepted and published:

1.The dataset used in the study contains 4,040 images with predefined augmentation techniques. This may limit the model's ability to generalize to diverse road conditions and surface types in real-world applications. A broader, more varied dataset could improve the model’s adaptability across different environments.

2.While the model demonstrates strong performance in controlled experiments, there is a lack of real-world validation to assess its robustness under different environmental conditions (e.g., varying lighting, weather, and road types). It is crucial to test the model in non-controlled, field-based scenarios to ensure its practical applicability.

3.Despite the focus on a lightweight model, the experimental setup relies on high-performance hardware such as the NVIDIA RTX 4090 GPU. This contradicts the paper’s claim of low computational cost, especially for deployment in mobile or resource-constrained environments. It would be helpful to see performance metrics on more typical, low-cost hardware configurations.

4.The paper compares the Flexi-YOLO model with YOLOv5, YOLOv7, and YOLOv8, but it does not include other state-of-the-art lightweight models (e.g., transformer-based models or more recent, efficient architectures). Including a broader range of models for comparison would provide a more comprehensive evaluation of the Flexi-YOLO model’s performance and efficiency.

5.The authors use several image augmentation techniques to improve model robustness. It would be beneficial to briefly discuss how these augmentation methods specifically influence the model’s performance, particularly for detecting cracks under varying conditions (e.g., different weather, lighting, and road types). A deeper analysis of the role and effectiveness of these augmentation strategies would offer valuable insights into how the model generalizes across real-world scenarios. For example, the ideas in these paper can be used as a reference:1.Automatic detection of tunnel lining crack based on mobile image acquisition system and deep learning ensemble model. https://doi.org/10.1016/j.tust.2024.106124. 2. Tunnel lining crack detection model based on improved YOLOv5. https://doi.org/10.1016/j.tust.2024.105713

The paper introduces several advanced techniques (e.g., GAM, AKConv) to enhance detection accuracy. However, it would be helpful to include a brief discussion on the interpretability of the model. How can users or practitioners understand and interpret the decisions made by the model, especially in edge cases where false positives or negatives occur? Providing insight into the model’s decision-making process could enhance its acceptance in real-world applications.

6. PLOS authors have the option to publish the peer review history of their article (what does this mean?). If published, this will include your full peer review and any attached files.

Reviewer #1: No

Reviewer #2: No

---

## [Author Response · Author response to Decision Letter 1]

13 Mar 2025

Response to Reviewers

Title: Flexi-YOLO: A Lightweight Method for Road Crack Detection in Complex Environments

Manuscript Number: PONE-D-24-58741

Author: Jiexiang Yang *,+, Renjie Tian +, Zexing Zhou , Xingyue Tan and Pingyang He

First of all, the authors would like to express their sincere thanks to the reviewers and editors for helpful comments and suggestions. The explanation of the modifications as well as corrections in this revision can be arranged as follows.

The comments of the reviewers are listed in [comment] in italics below. Our reply is in normal font in [reply]. The modified content is given in [modified] in normal font. The modification details are tracked in detail with the latexdiff given on the official website and named Revised Manuscript with Track Changes.pdf . The modifications or additions to the manuscript are marked in blue text.

Response to Journal Requirements

[Comment]1.Please ensure that your manuscript meets PLOS ONE's style requirements, including those for file naming. The PLOS ONE style templates can be found at https://journals.plos.org/plosone/s/file?id=wjVg/PLOSOne_formatting_sample_main_body.pdf and https://journals.plos.org/plosone/s/file?id=ba62/PLOSOne_formatting_sample_title_authors_affiliations.pdf

[Reply]

Thank you for your reminding. We have combined the templates in the official website to make the manuscript typesetting to ensure that the manuscript meets the style requirements of PLoS One. Thank you for your reminding of the manuscript format and file naming. If you have any suggestions, please contact us by email. Thank you.

[Comment] 2. PLOS requires an ORCID iD for the corresponding author in Editorial Manager on papers submitted after December 6th, 2016. Please ensure that you have an ORCID iD and that it is validated in Editorial Manager. To do this, go to ‘Update my Information’ (in the upper left-hand corner of the main menu), and click on the Fetch/Validate link next to the ORCID field. This will take you to the ORCID site and allow you to create a new iD or authenticate a pre-existing iD in Editorial Manager. [Reply]

Thank you for your reminding. We have verified ORCID in the Edit Manager and achieved this.

[Comment] 3. Please note that PLOS ONE has specific guidelines on code sharing for submissions in which author-generated code underpins the findings in the manuscript. In these cases, we expect all author-generated code to be made available without restrictions upon publication of the work. Please review our guidelines at https://journals.plos.org/plosone/s/materials-and-software-sharing#loc-sharing-code and ensure that your code is shared in a way that follows best practice and facilitates reproducibility and reuse.

[Reply]

Thank you for your reminder. We have uploaded the relevant code to GitHub during submission, and the repository is available at: https://github.com/Noceiling1/code.git. We hope that providing the complete code resources will assist the editors and reviewers in better evaluating the research work presented in our manuscript. Should you have any questions during the review process, we are available to provide further clarification via email at any time.

[Comment] 4. Thank you for stating the following financial disclosure:

This work was supported by the National Natural Science Foundation of China under Grants (62276034、62306052), Group Building Scientific Innovation Project for Universities in Chongqing (CXQT21021), Joint Training Base Construction Project for Graduate Students in Chongqing (JDLHPYJD2021016), and Science and Technology Research Program of Chongqing Municipal Education Commission (KJQN202100712). 

Please state what role the funders took in the study.  If the funders had no role, please state: "left"The funders had no role in study design, data collection and analysis, decision to publish, or preparation of the manuscript."left"

[Reply]

Thank you very much for the rigorous and meticulous inspection and requirements of the journal in terms of financial disclosure. For the sponsor's statement, "the sponsor has not participated in research design, data collection and analysis, decision to publish or write manuscripts, and the sponsor only supports the hardware equipment and computing support of the experiment and the payment of publishing layout fees."

[Comment] 5. We note that your Data Availability Statement is currently as follows: All relevant data are within the manuscript and its Supporting Information files.

[Reply]

Thank you very much for your reminding about the dataset. The data supporting the findings of this study are openly available in Zenodo at [https://doi.org/10.5281/zenodo.15019275]. For further inquiries, please contact the corresponding author.

[Comment] 6. We are unable to open your Supporting Information file Flexi-YOLO.tex. Please kindly revise as necessary and re-upload.

[Reply]

Sorry about the problem with tex files. We have updated the openable integrated tex file package.

Response to Reviewer #1

[Comment] Language Correction:

Abstract:

Rephrase for conciseness and impact. For instance:

"Road crack detection is vital for infrastructure maintenance and public safety, yet complex backgrounds and irregular crack shapes challenge real-time, efficient, and accurate detection. This paper proposes Flexi-YOLO, a lightweight and robust model based on YOLOv8, incorporating novel components to address these challenges. Key innovations include the Wise-IoU loss function for enhanced regression accuracy and robustness, the DCNv-C2f module for feature transformation, and the Global Attention Module (GAM) for improved global information perception. Flexi-YOLO achieves significant improvements in accuracy, recall, and mAP metrics, offering a cost-effective solution for automated crack detection."

[Reply]

Thank you very much for your suggestion. We have rephrased the summary to make it more concise and highlight the core impact.

[Modified]

Road crack detection is crucial for the maintenance of infrastructure and public safety. However, the challenges posed by complex background environments and irregular crack shapes demand real-time, efficient, and precise detection. This paper proposes a lightweight yet robust Flexi-YOLO model based on the YOLOv8 algorithm. We designed Wise-IoU as the model's loss function to optimize the regression accuracy of its bounding boxes and enhance robustness to low-quality samples. The DCNv-C2f module is constructed for the transformation and fusion of feature information, allowing the convolutional kernels to adapt to the complex shape characteristics of cracks dynamically. A Global Attention Module (GAM) is integrated to improve the model's perception of global information. The AKConv convolution operation is employed to adaptively adjust the size of convolutions, further enhancing local feature capturing. Additionally, a lightweight network design is implemented, establishing G-Head (Ghost-Head) as the detection head to optimize the issue of feature redundancy. Experimental results show that Flexi-YOLO achieves an accuracy increase of 2.7% over YOLOv8n, a recall rate rise of 4.7%, a mAP@0.5 improvement of 5.3%, a mAP@0.5-0.95 increase of 3.9%, a decrease of 0.5 in GFLOPS, and an F1 score improvement from 0.80 to 0.84. Flexi-YOLO offers higher detection accuracy and robustness and meets the industrial demands for lightweight real-time detection and lower application costs, providing an efficient and precise solution for the automated detection of road cracks.

[Comment] Introduction:

Rewrite to engage the reader and highlight the importance of the work globally, not just in a single country. Example:

"Road networks form the backbone of global infrastructure, critical to economic growth and societal well-being. However, environmental stressors and prolonged usage lead to structural deterioration, with cracks posing risks to safety, durability, and usability. Timely and accurate crack detection is essential to mitigate these issues, reduce maintenance costs, and ensure sustainability. This paper addresses these challenges by proposing an innovative, lightweight model tailored for real-time crack detection in diverse and complex environments."

[Reply]

Thank you very much for your comments. We restated the content of the introduction. According to your suggestions, we reflected the background and importance of our work on a global scale rather than just discussing one country.

[Modified]

Roads are a critical component of global infrastructure and are essential for the long-term stable development of the world economy. However, due to prolonged use and various environmental factors, infrastructure structures inevitably experience aging, with concrete structures particularly susceptible to cracking [1]. These cracks affect road durability, safety, and usability, reduce their load-bearing capacity, shorten their lifespan, and even threaten user safety [2].According to the World Health Organization's 2023 report, approximately 1.19 million people die each year due to road traffic accidents, with nearly 3,400 deaths per day and up to 50 million injured. Among these, individuals aged 5 to 29 represent the primary age group for road accident fatalities, which is also a crucial demographic for nations' current and future economic development, bearing the responsibility for significant economic construction.

[Comment] Related Work:

Ensure smooth transitions between traditional methods and deep learning advancements.

[Reply]

Thank you for your professional comments on our article. We have optimized the relevant work to make the transition between traditional methods and deep learning methods more smooth and natural.

[Modified]

In recent years, with the rapid development of computer vision, researchers have provided new technological directions for crack detection, which can be broadly divided into two categories. The first category is based on traditional digital image processing, primarily relying on manual feature discrimination. This involves designing specific feature recognition criteria to facilitate identification [15-17]. For instance, Kim J et al. (2003)[18] achieved crack detection and localization through structural frequency changes, Nguyen et al. (2014) [19] utilized the geometric properties of cracks in images to detect crack edges, and Jin et al. (2018) [20] proposed a method for pavement crack detection that integrates directional gradient histograms with watershed algorithms. However, traditional image processing methods struggle to meet crack detection's precision and time requirements [21]. The second category is based on deep learning theory, employing convolutional neural networks for autonomous feature learning, thus achieving target detection of cracks. Deep learning has emerged as a mainstream detection method in recent years, characterized by high automation, intelligence, accuracy, and robustness, and has been widely adopted by scholars [22-24]. Utilizing deep learning methods reduces detection errors caused by human subjectivity and addresses the limitations of traditional image processing methods in recognizing complex backgrounds in practical engineering contexts.

[Comment]Expand on limitations of existing YOLO-based methods and how Flexi-YOLO surpasses them.

[Reply]

Thank you for your concern about this aspect of our manuscript. We have added more literature review in the introduction based on the limitations and defects of Yolo algorithm, and briefly described the work content of this paper in the key part to highlight the advantages of Flexi Yolo.

[Modified]

In the field of crack detection, there are typically two stages: the first involves traditional detection methods. However, with the advancement of artificial intelligence, deep learning detection is gradually replacing traditional methods. While deep learning models have made significant progress in road crack detection, they still face several key challenges. Factors such as background noise, surface texture, lighting shadows, crack edges, and the cracks' shape, size, depth, and moisture often lead to false positives and missed detections, increasing the difficulty of detection [7]. These factors require models to not only adapt to different background noise but also possess the capability to handle disturbances from surface textures and shadows. Road safety work is often conducted using mobile devices, which typically have limited computational power and lower application costs. Therefore, it is essential to ensure the model's lightweight and flexibility to accommodate such devices. Two-stage detection algorithms based on deep learning generally require large datasets and high computational costs, while single-stage detection methods offer faster speeds and smaller sizes. With the development of the YOLO series, the models' inference speed and lightweight nature have been further enhanced. Comparing YOLOv8, YOLOv9, and YOLOv10, it is found that YOLOv8 maintains lower computational costs while ensuring accuracy and inference speed [8]. Although YOLOv8 demonstrates strong multi-scale feature extraction capabilities, its detection accuracy for small objects and defect cracks is relatively low, leading to missed and false detections. The main reason is that as the network depth increases, some shallow feature information becomes difficult to retain [9].

In response to the aforementioned issues, this paper proposes the Flexi-YOLO model, which is based on the YOLOv8 algorithm and tailored explicitly for road crack detection, effectively addressing current challenges in identifying road cracks. This work aims to overcome the difficulties in road crack detection by achieving the following objectives:

Develop a lightweight model suitable for deployment on devices with limited computational resources, design a backbone network that enhances the capability to extract features of complex geometrical shapes of cracks, integrate attention mechanisms and loss functions to improve the model's detection accuracy and robustness, and redesign a detection head to reduce feature redundancy and computational load, ensuring the model's lightweight nature to meet industrial requirements.

The following are the main points of discussion in this article:

1.Introducing variable convolution DCNv3 in the C2f module enhances the feature extraction of irregularly shaped cracks, which effectively solves the problem of difficu

---

## [Decision Letter · Decision Letter 1]

9 Apr 2025

PONE-D-24-58741R1Flexi-YOLO : A Lightweight Method for Road Crack Detection in Complex EnvironmentsPLOS ONE

Dear Dr. Yang,

Thank you for submitting your manuscript to PLOS ONE. After careful consideration, we feel that it has merit but does not fully meet PLOS ONE’s publication criteria as it currently stands. Therefore, we invite you to submit a revised version of the manuscript that addresses the points raised during the review process.

We look forward to receiving your revised manuscript.

Kind regards,

Peng Geng

Academic Editor

PLOS ONE

Additional Editor Comments (if provided):

It is recommended that authors carefully revise according to the reviewer's suggestions, otherwise the reviewer may not approve it.

Reviewers' comments:

Reviewer's Responses to Questions

**Comments to the Author**

1. If the authors have adequately addressed your comments raised in a previous round of review and you feel that this manuscript is now acceptable for publication, you may indicate that here to bypass the “Comments to the Author” section, enter your conflict of interest statement in the “Confidential to Editor” section, and submit your "Accept" recommendation.

Reviewer #1: (No Response)

Reviewer #2: All comments have been addressed

Reviewer #3: All comments have been addressed

2. Is the manuscript technically sound, and do the data support the conclusions?

Reviewer #1: Partly

Reviewer #2: Yes

Reviewer #3: Yes

3. Has the statistical analysis been performed appropriately and rigorously? 

Reviewer #1: No

Reviewer #2: Yes

Reviewer #3: N/A

4. Have the authors made all data underlying the findings in their manuscript fully available?

Reviewer #1: Yes

Reviewer #2: Yes

Reviewer #3: Yes

5. Is the manuscript presented in an intelligible fashion and written in standard English?

Reviewer #1: No

Reviewer #2: Yes

Reviewer #3: Yes

6. Review Comments to the Author

Reviewer #1: Please see carefully the comments and give reply. All the comments are not answered satisfactorily.Literature review should be strengthened and find the critical gap of literature.

10.1016/j.conbuildmat.2023.134212

10.1007/s11709-022-0819-z

Reviewer #2: I don't have any other comments. My suggestions have been fully considered and the article has met the requirements for publication.

Reviewer #3: The manuscript presents Flexi-YOLO, a lightweight architecture based on YOLOv8 specifically for road crack detection in complex and resource-constrained environments. The authors innovatively combined several advanced techniques to enhance detection accuracy while ensuring computational efficiency suitable for edge devices. The manuscript is generally well-structured, rich in methodological details, and complemented by thorough experiments. The authors not only clearly described data preprocessing methods but also explicitly defined the performance metrics, providing associated formulas, and presented results with abundant diagrams and tables that assists readers in interpreting the robustness and reliability of the proposed method and findings. However, there are still some improvements to further increase the academic rigor of the paper.

Minor comments

1. Please provide clear links and/or references to the source dataset Roboflow crack dataset for full credibility and reproducibility.

2. The section of “Model Generalization Experiment”, on page 24, is considerate and well-executed, however, adding additional details about the class distribution within the datasets would be helpful.

3. The section of “Computational Analysis of Low-Power Devices”, on page 26 and 27, mentions good performance on hardware test. Could you expand this section by including quantitative metrics such as power consumption, latency during inference, memory consumption, and frames per second? This guarantees the sustainability and suitability for edge deployment.

7. PLOS authors have the option to publish the peer review history of their article (what does this mean?). If published, this will include your full peer review and any attached files.

Reviewer #1: **Yes: **Amit Shiuly

Reviewer #2: No

Reviewer #3: No

---

## [Author Response · Author response to Decision Letter 2]

22 Apr 2025

Response to Reviewers

Title: Flexi-YOLO: A Lightweight Method for Road Crack Detection in Complex Environments

Manuscript Number: PONE-D-24-58741

Author: Jiexiang Yang *,+, Renjie Tian +, Zexing Zhou , Xingyue Tan and Pingyang He

First of all, the authors would like to express their sincere thanks to the reviewers and editors for helpful comments and suggestions. The explanation of the modifications as well as corrections in this revision can be arranged as follows.

The comments of the reviewers are listed in [comment] in italics below. Our reply is in normal font in [reply]. The modified content is given in [modified] in normal font. The modification details are tracked in detail with the latexdiff given on the official website and named Revised Manuscript with Track Changes.pdf . The modifications or additions to the manuscript are marked in blue text.

Response to Reviewer #1

We sincerely thank you for your thorough review and valuable suggestions regarding our article. After carefully studying and implementing each of your proposed revisions, we are pleased to find that the quality of the article has significantly improved from its initial version. Your professional comments have helped us avoid numerous shortcomings and research pitfalls, resulting in a more rigorous academic expression and a more coherent structure. We have made every effort to revise based on your feedback and hope that you will recognize our efforts.

Once again, we express our heartfelt gratitude; your insights will have profound implications for our future research endeavors.

[Comment]

Language Correction:

Please see carefully the comments and give reply. All the comments are not answered satisfactorily.Literature review should be strengthened and find the critical gap of literature.

[Reply]

Thank you for your valuable feedback on our manuscript. We deeply recognize the shortcomings in the critical discussion of the literature in our previous revisions, as well as our failure to adequately address the key issues you raised. We sincerely apologize for this. Consequently, we have systematically re-evaluated and revised the literature in response to the inappropriate modifications mentioned in your last review. We have placed special emphasis on your important suggestion to "enhance the literature review and identify critical gaps," and the revised content is presented in the following sections.

[Comment]

Language Correction:

Abstract:

Rephrase for conciseness and impact. For instance:

"Road crack detection is vital for infrastructure maintenance and public safety, yet complex backgrounds and irregular crack shapes challenge real-time, efficient, and accurate detection. This paper proposes Flexi-YOLO, a lightweight and robust model based on YOLOv8, incorporating novel components to address these challenges. Key innovations include the Wise-IoU loss function for enhanced regression accuracy and robustness, the DCNv-C2f module for feature transformation, and the Global Attention Module (GAM) for improved global information perception. Flexi-YOLO achieves significant improvements in accuracy, recall, and mAP metrics, offering a cost-effective solution for automated crack detection."

[Reply]

Thank you very much for your suggestion. We have rephrased the summary to make it more concise and highlight the core impact.

[Modified]

Road crack detection is critical to global infrastructure maintenance and public safety, and complex background environments and nonlinear damage crack patterns challenge the need for real-time, efficient, and accurate detection.This paper proposes a lightweight yet robust Flexi-YOLO model based on the YOLOv8 algorithm. We designed Wise-IoU as the model's loss function to optimize the regression accuracy of its bounding boxes and enhance robustness to low-quality samples. The DCNv-C2f module is constructed for the transformation and fusion of feature information, allowing the convolutional kernels to adapt to the complex shape characteristics of cracks dynamically. A Global Attention Module (GAM) is integrated to improve the model's perception of global information. The AKConv convolution operation is employed to adaptively adjust the size of convolutions, further enhancing local feature capturing. Additionally, a lightweight network design is implemented, establishing G-Head (Ghost-Head) as the detection head to optimize the issue of feature redundancy. Experimental results show that Flexi-YOLO achieves an accuracy increase of 2.7% over YOLOv8n, a recall rate rise of 4.7%, a mAP@0.5 improvement of 5.3%, a mAP@0.5-0.95 increase of 3.9%, a decrease of 0.5 in GFLOPS, and an F1 score improvement from 0.80 to 0.84. Flexi-YOLO offers higher detection accuracy and robustness and meets the industrial demands for lightweight real-time detection and lower application costs, providing an efficient and precise solution for the automated detection of road cracks.

[Comment]

Related Work:

Ensure smooth transitions between traditional methods and deep learning advancements.

[Reply]

Thank you for your professional comments on our manuscript. We have re-optimized the introduction and related work sections, adding several transitional elements to ensure a smoother and more natural flow between traditional methods and deep learning approaches.

[Modified]

(1) Introduction:

In the field of crack detection, it is usually divided into two stages. The first stage involves using traditional methods for detection. However, with the advancement of artificial intelligence, methods utilizing computer vision are gradually replacing traditional techniques. Although mainstream deep learning methods in road crack detection have made significant progress, they still face several key challenges. Factors such as background noise, surface texture, lighting shadows, crack edges, and variations in crack shape, size, depth, and moisture often lead to false positives and missed detections, increasing the difficulty of detection [9]. These factors require models to not only adapt to different background noises but also possess the ability to handle surface textures and shadows, enhancing their resistance to interference. Road safety work is often conducted on mobile devices that typically have limited computational power and lower application costs. Therefore, it is necessary to ensure that the models are lightweight and flexible enough to adapt to such devices.

(2) Related Work:

①Traditional methods:

Manual inspection and visual examination in crack detection remain the primary methods in most developing countries. However, manual inspection faces a series of issues, including but not limited to low efficiency, subjective judgment factors affecting assessments, difficulty in detecting fine cracks with the naked eye, high labor costs, and significant errors [10]. Thermal imaging technology has also garnered attention for crack detection due to its high portability and insensitivity to lighting conditions, offering specific advantages; however, this method is quite expensive, and its resolution needs improvement [11]. Ultraviolet and laser techniques [10][12]are frequently used in the detection of pavement cracks, as they can produce high-resolution images. However, this approach requires complex equipment and has high costs. Ground Penetrating Radar (GPR) is a non-destructive testing method widely used for pavement inspection [13][14], but it generates large amounts of data, has slow processing times, and suffers from subjective data interpretation and inconsistent standards. While these traditional techniques each have advantages, none have successfully balanced the issues of accuracy, efficiency, and cost.

In recent years, with the rapid development of computer vision, researchers have provided new technological directions for crack detection, which can be broadly categorized into two types. The first type is based on traditional digital image processing, primarily involving manual feature discrimination, where specific feature recognition criteria are designed to facilitate identification[15-17]. Kim J T et al. [18] utilized structural frequency changes to detect and locate cracks, while Nguyen et al. [19] employed the geometric properties of cracks in images to identify their edges. Jin et al. (2018) [20] proposed a method for detecting pavement cracks by integrating directional gradient histograms with the watershed algorithm. However, traditional image processing methods struggle to meet crack detection's accuracy and time requirements [21]. It is noteworthy that while traditional crack detection methods achieved some success compared to manual inspections in earlier stages, their heavy reliance on human intervention and fixed detection scenarios has led to persistent issues such as inefficiency and poor generalization performance, prompting researchers to explore alternative approaches. The second type is based on deep learning theory, utilizing convolutional neural networks for autonomous feature learning, thereby achieving the task of crack detection. Due to its high levels of automation, intelligence, precision, and robustness, this approach has become the mainstream method in the detection field and is widely adopted by scholars[22-24].

②Deep Learning Methods:

Deep learning methods have addressed some issues of traditional crack detection techniques, reducing detection errors caused by human subjectivity and overcoming the limitations of conventional image processing methods in recognizing complex backgrounds in practical engineering applications.

[Comment]

Expand on limitations of existing YOLO-based methods and how Flexi-YOLO surpasses them.

[Reply]

Thank you for your attention and thorough review of our manuscript. In response to the limitations and shortcomings of the YOLO algorithm, we have added a more comprehensive literature review at the end of the related work section to illustrate these limitations. Additionally, we have briefly described the content and innovations of our work to highlight the advantages of Flexi-Yolo.

[Modified]

Sapkota et al. [47] found that, when compared to YOLOv8, YOLOv9, and YOLOv10, YOLOv8 maintains accuracy and inference speed while achieving a lower computational cost. However, despite its strong performance in multi-scale feature extraction, YOLOv8 still struggles with detecting small objects and defect cracks, leading to frequent missed detections and false positives. The primary issue is that as the network layers increase, some shallow feature information becomes difficult to retain [48].

One-stage algorithms provide faster detection speeds in real-time scenarios but sacrifice some level of accuracy. The optimization efforts in the studies mentioned above mainly focus on the backbone and neck networks, overlooking the potential of the detection head in model optimization. While these works have achieved remarkable outcomes, there are still deficiencies in balancing the model for detection tasks in non-ideal backgrounds and the pursuit of lightweight solutions. To address the limitations of deep learning methods in lightweight crack detection tasks under complex backgrounds, we aim to enhance the model's feature extraction and fusion capabilities by redesigning the backbone and neck network structures. Additionally, we propose an innovative detection head to fill the gap in optimizing the detection head to achieve model lightweightness, thereby catering to the adaptability of edge computing or mobile infrastructures.

[Comment]

More Literature Review:

Discuss advancements in deep learning for infrastructure health monitoring, such as applications beyond road cracks (e.g., bridge, pipeline, and building inspections).

[Reply]

We believe this is a well-considered and professional opinion. Based on your feedback, we have carefully reviewed the relevant literature and elaborated on the applications of crack detection in other fields within the related work section. By examining crack detection methods from other disciplines, we have made our manuscript more comprehensive and persuasive.

[Modified]

Cha et al.[32] used CNN in combination with the sliding window technique to scan the test images to make the model excel at detecting thin cracks under illumination conditions. Hacıefendioğlu et al.[33] used Faster R-CNN to study crack detection to conclude that lighting conditions have the highest impact on crack detection. Xu et al.[34] found that the former is better than the latter under certain conditions by comparing the detection of cracks with Faster R-CNN and Mask R-CNN, but the two algorithms require large and complex datasets. To address these problems, researchers have proposed various optimization strategies. For example, Zhang et al.[35] introduced MobileNetV2 and CBAM into YOLOv3 to reduce the network parameters. Yu et al.[36] pruned YOLOv4 to significantly increase the detection rate while ensuring the accuracy to contribute to the UAV crack detection. Chen et al.[37] introduced the Ghost module and CA attention mechanism in YOLOv5 for the backbone network to effectively improve the detection of coal cracks, which contributes to the construction of intelligent mines. Chen et al.[38] proposed a sample enhancement strategy based on cycle-GAN, which uses a migration learning strategy to incorporate the Transformer attention mechanism in YOLO v5 to help the detection of cracks by drones. Transformer attention mechanism, to help the network find the region of interest in the complex background, to improve the detection and identification of small-scale defects, for the pipeline defects task to solve the overfitting problem brought by small samples and large models. Tran et al.[39] compared the mainstream models, selected YOLOv7, and proposed a U-Net model that can detect and segment bridge deck cracks in a fast and high-precision way. detection and segmentation of bridge surface cracks. Xiong et al.[40] incorporated the GAM attention mechanism and Wise-IoU into YOLOv8 to study the bridge surface cracks and found that the GAM attention mechanism has a great effect on crack detection. Wang et al.[41] optimized YOLOv8 with simSPPF using the spatial pyramid pooling layer in YOLOv8 and introduced the dynamic large convolutional kernel LSK attention mechanism, making the model more effective for pavement defect detection. Liu et al. [42]introduced the C2f_AK module in the neck network to enhance feature fusion, significantly improved the detection head, and reduced parameters to achieve model lightweighting.

[Comment]Include recent works utilizing YOLO-based architectures in different domains to position Flexi-YOLO in the broader context.

[Reply]

Thank you very much for your suggestions; they have been immensely helpful for our manuscript. Your insights have broadened our perspective on the applications of YOLO technology across various fields, allowing our article to extend beyond just crack detection. Following the style of the reference papers you provided, we have included more recent literature in the related work section to further explore the applications of YOLO in different domains.

[Modified]

It is noteworthy that YOLO has not only achieved results in the field of crack detection but also in agricultural applications. Di et al. [43] proposed the DF-Tiny-YOLO model for apple leaf disease detection, effectively addressing the issue of distinguishing between healthy and diseased apple leaves. Liu et al. [44] introduced the ETSR-YOLO model to tackle the challenge of traffic sign recognition in road environments, providing a new solution for traffic sign recognition on embedded platforms in vehicles. Gong et al. [45] developed a two-stage ship detection method based on YOLOv8 for long-distance vessel detection, offering new insights into the field. Xu et al. [46] designed three plug-and-play modules—DGLFG, SCCFF, and ISPP—based on YOLOv8, significantly improving the feature learning capabilities for detecting small objects from aerial photography.

[Comment]

Highlight shortcomings of current methods, like high computational requirements, sensitivity to noise, or lack of adaptability to varying crack shapes.

[Reply]

Thank you for your suggestion. Based on the modifications we mad

---

## [Decision Letter · Decision Letter 2]

23 May 2025

Flexi-YOLO : A Lightweight Method for Road Crack Detection in Complex Environments

PONE-D-24-58741R2

Dear Dr. Yang,

We’re pleased to inform you that your manuscript has been judged scientifically suitable for publication and will be formally accepted for publication once it meets all outstanding technical requirements.

Kind regards,

Peng Geng

Academic Editor

PLOS ONE

Additional Editor Comments (optional):

Reviewers' comments:

Reviewer's Responses to Questions

**Comments to the Author**

1. If the authors have adequately addressed your comments raised in a previous round of review and you feel that this manuscript is now acceptable for publication, you may indicate that here to bypass the “Comments to the Author” section, enter your conflict of interest statement in the “Confidential to Editor” section, and submit your "Accept" recommendation.

Reviewer #1: All comments have been addressed

Reviewer #3: All comments have been addressed

2. Is the manuscript technically sound, and do the data support the conclusions?

Reviewer #1: Yes

Reviewer #3: Yes

3. Has the statistical analysis been performed appropriately and rigorously? 

Reviewer #1: Yes

Reviewer #3: Yes

4. Have the authors made all data underlying the findings in their manuscript fully available?

Reviewer #1: Yes

Reviewer #3: Yes

5. Is the manuscript presented in an intelligible fashion and written in standard English?

Reviewer #1: Yes

Reviewer #3: Yes

6. Review Comments to the Author

Reviewer #1: As the authors have addressed all the comments and concerns raised during the review process satisfactorily, I find no remaining issues. The manuscript meets the necessary standards and can be accepted for publication. I have no concerns regarding dual publication, research ethics, or publication ethics.

Reviewer #3: Thanks the authors for meticulously addressing my comments around providing more concrete quantitative metrics for the model performance, describing data distributions in the dataset, and including links/references to the dataset. I think the current manuscript is ready for publication.

7. PLOS authors have the option to publish the peer review history of their article (what does this mean?). If published, this will include your full peer review and any attached files.

Reviewer #1: **Yes: **Amit Shiuly

Reviewer #3: No

---

## [Editor Report · Acceptance letter]

PONE-D-24-58741R2

PLOS ONE

Dear Dr. Yang,

I'm pleased to inform you that your manuscript has been deemed suitable for publication in PLOS ONE. Congratulations! Your manuscript is now being handed over to our production team.

Kind regards,

on behalf of

Dr. Peng Geng

Academic Editor

PLOS ONE